# Mapping cumulative impacts to coastal ecosystem services in British Columbia

**Gerald G. Singh**[1,2¤*], **Ian M. S. Eddy**[3], **Benjamin S. Halpern**[4,5], **Rabin Neslo**[6], **Terre Satterfield**[2], **Kai M. A. Chan**[2]

**1** NEREUS Program, Institute for the Oceans and Fisheries, The University of British Columbia, Vancouver, British Columbia, Canada, **2** Institute for Resources, Environment, and Sustainability, The University of British Columbia, Vancouver, British Columbia, Canada, **3** Forest and Conservation Sciences, The University of British Columbia, Vancouver, British Columbia, Canada, **4** National Center for Ecological Analysis and Synthesis, Santa Barbara, California, United States of America, **5** Bren School of Environmental Science and Management, University of California, Santa Barbara, California, United States of America, **6** Julius Centre for Health Sciences and Primary Care, University Medical Centre Utrecht, Utrecht, The Netherlands

¤ Current address: Department of Geography, Memorial University, St. John's, Newfoundland, Canada
* geralds@mun.ca

**Data Availability Statement:** All relevant data are uploaded to Figshare: https://figshare.com/projects/Mapping_Cumulative_Impacts_to_Coastal_Ecosystem_Services_in_British_Columbia/79074.

## Abstract

Ecosystem services are impacted through restricting service supply, through limiting people from accessing services, and by affecting the quality of services. We map cumulative impacts to 8 different ecosystem services in coastal British Columbia using InVEST models, spatial data, and expert elicitation to quantify risk to each service from anthropogenic activities. We find that impact to service access and quality as well as impact to service supply results in greater severity of impact and a greater diversity of causal processes of impact than only considering impact to service supply. This suggests that limiting access to services and impacts to service quality may be important and understanding these kinds of impacts may complement our knowledge of impacts to biophysical systems that produce services. Some ecosystem services are at greater risk from climate stressors while others face greater risk from local activities. Prominent causal pathways of impact include limiting access and affecting quality. Mapping cumulative impacts to ecosystem services can yield rich insights, including highlighting areas of high impact and understanding causes of impact, and should be an essential management tool to help maintain the flow of services we benefit from.

## 1. Introduction

Humanity's great and growing influence on the planet demands an increased understanding of how multiple activities cumulatively affect the human benefits and values associated with the environment [1,2]. The need to understand and manage simultaneous impacts of multiple human activities on ecosystems (such as fisheries and agricultural runoff impacting fish habitat concurrently), referred to here as cumulative impacts, has led to widespread uptake in cumulative impact mapping methods around the world [3–11]. However, impact mapping studies generally reflect how human activities affect species and habitats, neglecting thus far how multiple activities cumulatively affect ecosystem services–the processes by which nature renders

**Funding:** GGS recieved funding from the Natural Sciences and Engineering Research Council of Canada and the Pacific Institute for Climate Solutions. KMAC recieved funding from the David and Lucile Packard Foundation for this work. The funders had no role in study design, data collection and analysis, decision to publish, or preparation of the manuscript.

**Competing interests:** The authors have declared that no competing interests exist.

benefits for people [12]. Understanding impacts on ecosystem services would allow for a representation of multiple societal benefits from the environment, enabling targeted management on specific ecosystem services. Assessments of impacts on ecosystem services could allow us to establish baseline knowledge of the ecosystem services and geographic areas facing the greatest impact, as well as help evaluate and plan for emerging impacts from local and global stressors (such as from future oil spills and climate change, respectfully).

Ecosystem services are the environmental processes that render benefits to people. Implicit to this definition is that, while ecosystem functions are essential for providing ecosystem services, these services do not exist without human beneficiaries [13,14]. Any human activity that impacts ecosystems has the potential to impact ecosystem services in multiple ways. In addition to impacts on the biophysical production of services, human activities and infrastructure can also undermine the "consumption" of ecosystem services [14].That is, a human activity can undermine people's ability to access or enjoy an ecosystem service. The role of impacts to the production versus the consumption of ecosystem services is largely unexplored in the literature, however. For example, in New Zealand shellfish aquaculture sites and shipping lanes can limit commercial fishing operations in an area because of legislation that limits their overlap, impacting the contribution of fisheries ecosystem services [15]. In this case, the assessed impact of shipping and aquaculture on fisheries operated through changes in access and not through impacts on biophysical supply (though the effluent from increased shipping may impact biophysical supply in the long term).

Various human activities and stressors (which we collectively called drivers) impact ecosystem services. We define drivers as the human activities and long-term stressors (such as ocean acidification) that contribute to a deterioration of benefits derived from ecosystem services. We define stressors as the processes that undermine ecosystem service benefits, and we define impacts as the deterioration of ecosystem service benefit. For example, agriculture contributes to runoff that can lead to sedimentation which can smother shellfish harvested by people [15,16]. In this example, agriculture is a driver, sedimentation is a stressor, and reduced shellfish biomass for food is the impact [15,17]. Impacts to ecosystem services can be characterized at each step in the ecosystem services 'cascade' [13], with impact drivers potentially affecting supply (the biophysical components that produce ecosystem services), service (the ability of people to access and benefit from a service), and value (people's preferences for ecosystem services, 13). Reframing the previous example, shipping lanes and aquaculture sites impact the service (the ability of people to access fisheries for food through legal restriction), even if the growth and availability of fish (the supply) might be unaffected. In this case (where shipping lanes and aquaculture restrict fisheries) what might not be considered an environmental impact (to fish) would be considered an ecosystem service impact. While this cascade is useful for parsing out the dynamics of impacts to ecosystem services, the relative importance of these factors (supply, service, and value) in regulating impact to ecosystem services is not known.

The ecosystem service cascade has potential repercussions for impact mapping. Because delivery of ecosystem services to people requires both the provision of services through biophysical means (the supply) and delivery to people (service) that demand those services (value), maps of ecosystem services may be more restricted in space than maps of total service supply [14,18,19]. The few existing studies mapping cumulative impacts to ecosystem services do so using human use and landscape proxies of ecosystem services [11,20]. Recently, spatial models have been created that utilize production functions for ecosystem services, relating landscape features important for ecosystem services, as well as spatial social data on human use of the environment, to generate maps of ecosystem services on the coast [21,22]. These models provide a more precise and comprehensive mapping of ecosystem services, including those without close human use proxies.

Beyond spatial representation, the ecosystem service cascade influences the metrics we use to measure impact. Existing frameworks of impact to ecosystem services characterize change in the underlying ecosystem as the principal (and sometimes sole) driver of impact, with human beneficiaries of services largely subject to changes in ecosystem service supply [23–26]. However, as mentioned previously, considering only the underlying ecosystem represents changes to potential ecosystem services without incorporating considerations of if and how humans use them. Including metrics that consider impacts to the service and value of ecosystem services can modify our understanding and measurements of impact. Maps of species and habitat may be sufficient to approximate impacts to ecosystem services when the mechanisms of impact operate mostly through biophysical supply. However, changes to people's access, use, and perceived quality of service may also be important for understanding impacts to ecosystem services [27,28], and understanding the mechanism of impacts on ecosystem services can help address management goals [17].

Here we model human impacts to specific ecosystem services on coastal British Columbia to identify areas of high impact in consideration of the ecosystem service cascade. In mapping impacts according to the ecosystem service cascade we also attempt to advance the understanding of impacts to ecosystem services. Coastal British Columbia is an area renowned for its scenery and productivity, contributing greatly to the economy, sense of place and other values important to residents and visitors [29,30]. Maps of cumulative impacts to coastal British Columbia ecosystems have been produced [8,10,31]. This work, alternately, does so for ecosystem services themselves, representing cumulative impact as the combined total impact that an ecosystem service experiences from a variety of co-occurring drivers of impact (such as ocean acidification, agricultural runoff, and fishing). We ask: 1) Which ecosystem services face the most severe cumulative impact in coastal British Columbia?; 2) What drivers pose the greatest threat to what ecosystem services?; 3) Where are ecosystem services under greatest threat?; 4) How do the answers to the first three questions change if measures of service and value are considered or left out (i.e. consider impacts to the ecosystem service cascade versus only to supply)?; 5) How may projected future impacts affect ecosystem services?; 6) What is the relative importance of metrics of service supply, service, and value to impacts on ecosystem services?; 7) What are the main causal pathways of impacts that affect the ecosystem services? Together, addressing these questions builds on established methods to map cumulative impacts using geospatial data and expert derived estimates of ecological vulnerability [4,32].

## 2. Methods

### 2.1 Study site

The coast of British Columbia, Canada spans a distance of almost 1000 km, with a complex shoreline geography of fjords, inlets, and islands extending over 25,000 km in length. It is a region of diverse resource harvesting important for ecological, economic and cultural reasons, many of which are unique to the region; for example glass sponge reefs, globally significant seabird populations, salmon, eulachon, and resident orca. The region is also important culturally for intangible benefits, including nature-based tourism. A broad range of human activities occur in this region, and a multiple cumulative impact studies have been conducted to assess impacts on the marine ecosystems [8,10,28]. Sea-based activities include fishing, aquaculture, tourism, utility and transportation. Coastal activities also influence the marine and estuarine resources in this region, including human settlement, ports and marinas, and log storage and handling. Land-based activities occurring in the watersheds are connected to coastal marine systems through freshwater runoff and include forestry, agriculture, mining and pulp and paper mills. The region is also subject to impacts from long-range and global stressors such as

climate change, pollutants and debris. Activities that include vessel use additionally include the stressors associated with either small or large vessel use in their cumulative risk. Management of coastal British Columbia is done in a piecemeal way (often with little coordination between regulators), with sea-based activities under the purview of Fisheries and Oceans Canada, land-based resources under provincial authority (Forest, Lands, Natural Resource Operations and Rural Development), coastal national parks under Parks Canada, and Environment and Climate Change Canada, and towns and human settlements often governed by local governments. Because of the diverse natural resources and ecosystem services, as well as the past research done on ecological impacts, we chose to study this region as a case study to study cumulative impacts on ecosystem services.

### 2.2 Methodological overview

Mapping and quantifying impacts to ecosystem services requires both (1) understanding the location and intensity of drivers co-occurring with ecosystem services (i.e., the 'footprint' of drivers) and (2) the risk (the potential of a driver to impact an ecosystem service where they co-occur) each activity poses to each ecosystem service [23,24]. Measuring impact as a product of risk and the co-occurrence of activities (with measured intensities) and ecosystem services follows the conceptual structure of cumulative impact mapping [33], as diagrammed in Fig 1.

The analysis consisted of four main steps. 1) Spatial representation of ecosystem services: we mapped eight ecosystem services using InVEST models and spatial data available for the region. 2) Spatial representation of drivers of impact: we assembled spatial data for 21 drivers that potentially impact ecosystem services. 3) Risk assessment: we derived risk scores for each service-driver (risk of driver $x$ on service $y$) combination under current conditions (within the last 10 years) via an expert elicitation process. Ethics approval for this expert elicitation was

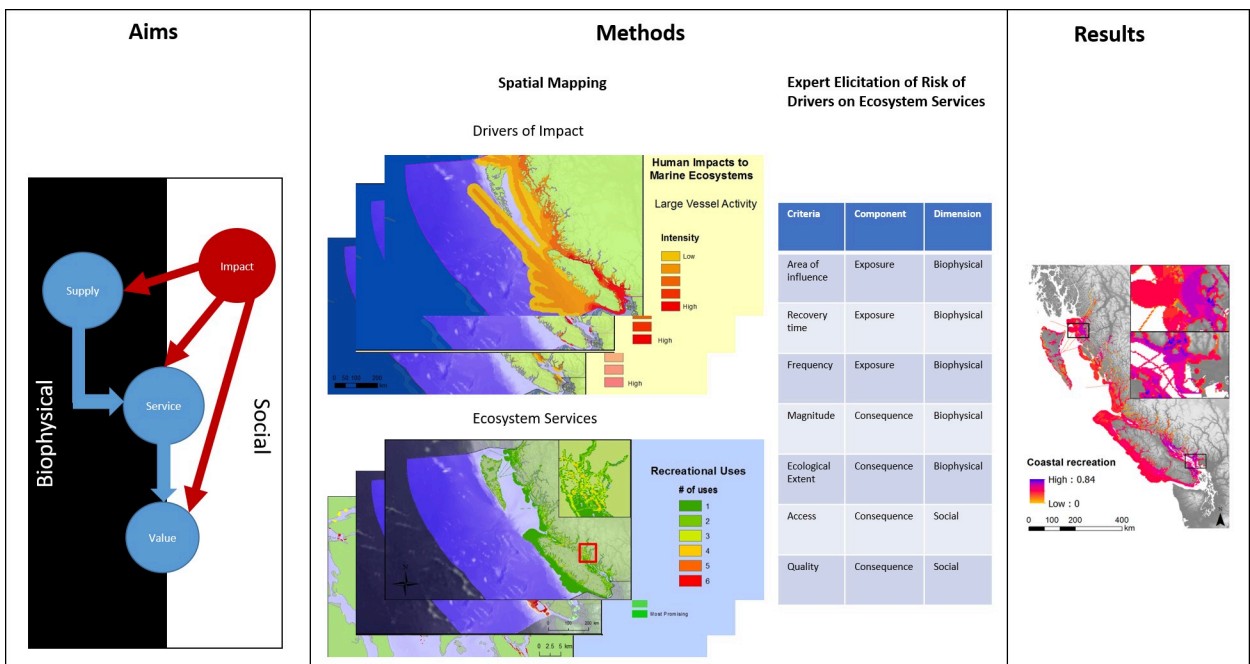

**Fig 1. The aims of the study are to assess impact to ecosystem services across the ecosystem service cascade of supply, service, and value dimensions of ecosystem services.** To do so, we employ the methods of cumulative impact mapping, which combines spatial data on multiple drivers of impact, multiple ecosystem services, and expert-derived estimates of the risk posed by drivers to ecosystem services, according to various risk criteria. As a result, we generate maps of cumulative impact on ecosystem services.

given by UBC's Behavioural Research Ethics Board (approval number H12-01868). Written consent was obtained from each participant to partake in the study. These risk scores were calculated by expert derived estimates of risk criteria and criteria weights, then combined with data on human activities and stressors generates impact scores. We also used expert elicitation to estimate the risk of key climate change and potential oil spills to ecosystem services in the future. To further explore how ecosystem services across supply, service and value, we asked experts to detail the causal pathways of impacts to ecosystem services. 4) Cumulative impacts model: we overlaid maps of drivers of impact (with impact scores) on maps of ecosystem service to assess the cumulative impacts of all available activities on each service, in accordance with our definition of cumulative impacts. The resulting maps allowed us to answer where ecosystem services were under greatest impact, which ecosystem services were most impacted, and by what driver. The expert scores allowed us to distinguish impacts on ecosystem service supply from impacts on service and value considerations. We compared maps of total impact with maps that only incorporated impacts on ecosystem service supply to explore the importance of service and value dimensions of ecosystem service risk. We detail each step below.

## 2.3 Spatial representation of ecosystem services

We mapped eight different ecosystem services using InVEST, including coastal aesthetics, coastal protection, benefits from commercial demersal fisheries, benefits from commercial pelagic fisheries, and coastal recreation [21,22], though for potential tidal and wave energy and benefits from aquaculture we used publicly available spatial data on the extents of these ecosystem services and did not need to use InVEST (see S2 Table). InVEST (integrated valuation of ecosystem services and tradeoffs) is a decision-support tool for mapping and valuing ecosystem services, that generates spatially explicit models of ecosystem services based on underlying ecosystem characteristics. Coastal aesthetics was modeled by calculating the viewshed from kayaking, recreational boating, population centers, recreational fishing. Coastal protection was modeled assessing the protection provided by marine vegetation (kelp and seagrass) to different types of shoreline (sandy to rocky). Benefits from commercial demersal fisheries and benefits from commercial pelagic fisheries were modeled by aggregating multiple commercial fishery spatial data layers. Coastal recreation includes kayak, recreational boating, recreational fishing, and populous sites for recreation, including camping and dive sites. We modeled "potential" energy generation because British Columbia currently does not have wave and tidal energy operations, but there is interest in harnessing this energy supply. Benefits from finfish and shellfish aquaculture were modeled by aggregating spatial data of finfish and shellfish aquaculture. For more detail on the ecosystem service models, see S1 File.

The InVEST tool has tiered models for mapping ecosystem services based on different levels of data availability. The highest tier InVEST models are capable of quantifying and calculating monetary values of ecosystem services within the area that people use them [17]. Due to data limitations, we were prevented from modeling ecosystem services at the most refined tier, but we could produce maps of the extent of human use of ecosystem services for all eight across coastal BC. We used the base InVEST models for fisheries and recreation maps whereby overlapping maps of different activities creates the resulting service model. We modeled coastal aesthetics with InVEST by calculating the viewshed from sites of recreation and human habitation. This model considers topography and the curvature of the earth to calculate the viewshed. We modeled coastal protection with InVEST by mapping the parts of the coast protected by vegetation, kelp, and erosion-resistant substrate (not mapped are areas of the coast without protection). We did not use InVEST to map potential renewable energy and benefits from aquaculture, as we opted to use instead the publicly available spatial data on wave and

tidal energy areas of interest along the BC coast, as well as the locations of shellfish and finfish aquaculture. See the Appendix A for detailed descriptions of ecosystem service model parameterization.

## 2.4 Spatial representation of drivers of impact

We assembled spatial data layers for 21 different drivers of impact (processes that impact the environment from either human activities or long term change), including drivers related to fisheries, coastal commercial industries, land conversion and management, and climate change impacts (these broad categories derived from [8], see Table 1). These spatial data layers included the spatial range of each driver, as well as the intensity of each activity within its range (for example, how many ships were using a particular shipping lane). Many human activities, such as fishing, access benefits from ecological processes and play important roles in

**Table 1. The ecosystem services modeled in our study and all associated human activities and stressors that pose risk to these ecosystem services.**

| Ecosystem Service | Human activity or stressor causing impact |
|---|---|
| Coastal Aesthetics | demersal destructive fishing; demersal non-destructive low bycatch fishing; demersal non-destructive high bycatch fishing; pelagic low bycatch fishing; pelagic high bycatch fishing; recreational fishing; finfish aquaculture; shellfish aquaculture; large boat traffic; ports, marinas, and harbours; small docks, ramps, wharfs; log dumping, handling, storage; ocean dumping; industry; pulp and paper; onshore mining; human settlements; agriculture |
| Coastal Protection | recreational fishing; large boat traffic; ports, marinas, and harbours; small docks, ramps, wharfs; log dumping, handling, storage; industry; pulp and paper; onshore mining; human settlements; agriculture; sea level rise |
| Benefits from Commercial Demersal Fishing | demersal destructive fishing; demersal non-destructive low bycatch fishing; pelagic low bycatch fishing; pelagic high bycatch fishing; recreational fishing; finfish aquaculture; shellfish aquaculture; large boat traffic; ports, marinas, and harbours; small docks, ramps, wharfs; log dumping, handling, storage; ocean dumping; industry; pulp and paper; onshore mining; human settlements; agriculture; ocean acidification; sea temperature change; UV change |
| Benefits from Commercial Pelagic Fishing | demersal destructive fishing; demersal non-destructive low bycatch fishing; pelagic low bycatch fishing; pelagic high bycatch fishing; recreational fishing; finfish aquaculture; shellfish aquaculture; large boat traffic; ports, marinas, and harbours; small docks, ramps, wharfs; log dumping, handling, storage; ocean dumping; industry; pulp and paper; onshore mining; human settlements; agriculture; ocean acidification; sea temperature change; UV change |
| Coastal Recreation | demersal destructive fishing; demersal non-destructive low bycatch fishing; demersal non-destructive high bycatch fishing; pelagic low bycatch fishing; pelagic high bycatch fishing; recreational fishing; finfish aquaculture; shellfish aquaculture; large boat traffic; ports, marinas, and harbours; small docks, ramps, wharfs; log dumping, handling, storage; ocean dumping; industry; pulp and paper; onshore mining; human settlements; agriculture; ocean acidification; sea level rise; sea temperature change; UV change |
| Potential Energy Generation | demersal destructive fishing; demersal non-destructive low bycatch fishing; demersal non-destructive high bycatch fishing; pelagic low bycatch fishing; pelagic high bycatch fishing; recreational fishing; large boat traffic; ports, marinas, and harbours; ocean dumping; industry |
| Benefits from Finfish Aquaculture | demersal destructive fishing; demersal non-destructive low bycatch fishing; pelagic low bycatch fishing; finfish aquaculture; large boat traffic; ports, marinas, and harbours; industry; pulp and paper; onshore mining; human settlements; ocean acidification; sea temperature change; UV change |
| Benefits from Shellfish Aquaculture | demersal destructive fishing; demersal non-destructive low bycatch fishing; pelagic low bycatch fishing; shellfish aquaculture; large boat traffic; ports, marinas, and harbours; small docks, ramps, wharfs; pulp and paper; onshore mining; human settlements; ocean acidification; sea level rise; sea temperature change; UV change |

ecosystem service delivery to people while also contributing impacts towards ecosystem services [15]. We treat these activities (e.g. fishing), therefore as both ecosystem services as well as drivers that cause impact (following Singh et al. [15]). To distinguish between these multiple roles that fisheries play, we emphasize benefits when labeling fisheries as ecosystem services (such as "benefits from commercial demersal fisheries") and emphasize impacts when labeling fisheries as drivers of impact (such as "demersal destructive fishing"). We treat ecosystem services as broad categories (such as demersal vs pelagic fisheries) and drivers that cause impact as specific categories because experts indicated that broad types of ecosystem services (such as various benthic fisheries, or various pelagic fisheries) are impacted in similar ways, while they indicated that they did not treat human activities and stressors in a similar way. Many of the data layers of drivers that cause impact were adapted from a previous cumulative impact study by Ban et al. [8] supplemented with data from British Columbia Marine Conservation Analysis [33] and GeoBC [34]. We compiled the 25 fisheries used by Ban et al. [8] into five categories (demersal destructive, demersal non-destructive, pelagic low bycatch, pelagic high bycatch, and recreational fishing) of fisheries that cause impact because the number of data layers influences the overall cumulative impact scores [8], and we did not want to overly bias impact based on fisheries scores. This dataset considers the area of influence of each human activity, with the extent of each area of influence dependent on prominent stressors (processes that cause impact) associated with each activity. We also included current climate stressors adapted and updated from Halpern et al. [35] global map (see S2 Table for data sources).

## 2.5 Risk assessment

Following the cumulative impact mapping approach first demonstrated by Halpern et al. [33], we overlay maps of impacting activities on ecosystem services, and calculate impact of activities by combining the spatial data of activity intensity with a measure of risk from a standard unit of human activity to a given ecosystem service. We define risk as the potential of a driver to impact a particular ecosystem service. In the context of our cumulative impact model, risk is the potential of a single event of an activity to impact a given ecosystem service. Calculating the quantitative estimates of risk and the model of cumulative impact were adapted from Halpern et al. [33] and are described in detail in the Supplementary Methods. Below we describe the expert elicitation process used to generate risk scores and summarize the cumulative impacts model that build from the ecosystem service maps, human activity maps, and the risk assessment.

**2.5.1 Expert elicitation for risk scores.** Our quantitative estimates of risk represent the potential impact that a unit of a driverposes to an ecosystem service when a driver co-occurs with an ecosystem service. To calculate the risk scores we relied on expert judgement, due to pervasive data gaps (see S3 Table for a description of drivers for risk quantification). We adapted the mail-in and phone expert survey used in Teck et al. [32] used to quantify ecosystem vulnerability to different drivers through ranking and quantification exercises, and adapted it for ecosystem services (survey description below). We used an online survey because it allowed us to reach all experts using a common platform. The diversity of ecosystem services and the large number of risk values precluded individual surveys, workshops, and other elicitation methods [36]. We invited a total of 437 experts to take part in the full survey (quantifying risk criteria, future risk criteria, generating risk criteria weights, and outlining mechanistic pathways of impacts), but 217 did not respond to the survey invitation, resulting in 220 potential expert responses (we could not determine whether these were appropriate experts who chose not to respond or if they did not receive or see the invitation). Of the resulting 220, 112 self-indicated that their level of expertise was not sufficient to quantify risk though

all 220 did provide responses on the mechanisms of impact. After accounting for non-responses and self-identification, we were left with a pool of 108 confirmed potential experts. Of this pool, 44 provided quantitative results on the survey (a 40.7% response rate).

Experts were selected by reviewing the literature of the various chosen ecosystem services in British Columbia and identifying authors of relevant studies. Authors and studies were identified through ISI Web of Knowledge with a focus on recruiting experts with subject-expertise in specific (or multiple) ecosystem services specifically within BC. We allowed participants to self-organize for chosen ecosystem services (some indicating their expertise for multiple ecosystem services), and they provided responses for all ecosystem services they presumed themselves experts on in BC. Risk estimates were compiled for benefits from commercial fisheries generally (instead of benefits from demersal and pelagic commercial fisheries separately), and we elicited risk scores for benefits from commercial aquaculture generally (instead of benefits from shellfish and finfish aquaculture specifically) because the fisheries and aquaculture experts indicated their expertise pertained to these ecosystem services across their subcategories.

Experts were tasked with quantifying risk according to seven criteria, building on those used in Teck et al. [29]. The criteria encompassed exposure (area of influence, frequency of impact and recovery time, S3 Table) and consequence (magnitude of impact on ecosystem service production, ecological extent of impact, effects to access and effects to perceived quality, S4 Table). The consequence criteria include considerations across the ecosystem service cascade. Impacts to supply dimensions are represented by magnitude of impact on service production and ecological extent, impacts to service dimensions are represented by effects to access, and impacts to value dimensions are represented by effects to perceived quality. Experts were instructed to consider current risk of activities to ecosystem services (within the last 10 years). For potential energy generation, only one expert provided these quantitative measurements (though others provided other information on potential energy generation) so quantitative results for this ecosystem service should be considered tentative, and future research should be taken to verify findings here. For all other ecosystem services, there were ≥3 experts providing measurements, consistent with expert input on previous cumulative mapping studies [4,29,35]. While we acknowledge that expert input can carry high uncertainty, expert input was the best option present give that no empirical results exist as an alternative, though empirically quantifying impacts to marine systems is a priority research area [37]. Despite this limitation, there is an established literature on using expert responses to inform decisions in contexts of limited data, and the particular expert-based approach used in cumulative impact mapping was evaluated in Teck et al. [29] and shown to be robust. Specifically for our study, risk criteria scores had relatively low variation across experts (standard deviation was usually less than half of the mean, and often less than a quarter of the mean for recreation, all fisheries, and all aquaculture). Additionally, experts were provided opportunities to comment and disagree with aggregated results, but in all ecosystem services, experts were satisfied with the results. Taken together (low variation across experts and no further refinement by experts), these results indicate that expert scores were relatively stable. See S6 Table for a summary of expert scores for the seven criteria across impacting activities for each ecosystem service.

**2.5.2 Future risk to ecosystem services.** To partially assess future risks to ecosystem services, experts were asked to quantify risk to two global driver and one regional driver of high concern, given the changing climate and development trajectory of British Columbia. These measures of future risk were not included in the final cumulative impact maps, as the maps only included risk estimates for current activities and stressors that cause impact. Experts were asked to quantify risk from sea surface temperature rise and ocean acidification according to projections for the year 2100 (3˚C increase and 0.3 pH decrease, respectively, [38] and to

quantify risk from a major oil spill ($>40\,000$ m$^3$, [39]). All risk scores were normalized so that the resulting expert scores were scaled between 0–1.

**2.5.3 Understanding mechanisms of impact.** We asked experts in the risk survey to indicate whether or not the given drivers of impact affected their chosen ecosystem service directly or indirectly (or neither or both), with an optional follow-up to describe the mechanism of impact. Each driver of impact were grouped in one of four different categories: fisheries impacts, coastal commercial impacts, land-based impacts, and climate change impacts. Fisheries impacts includes all those drivers related to fisheries including demersal destructive and non-destructive fishing, pelagic fishing, and recreational fishing. Coastal commercial impacts include coastal industries such as aquaculture, shipping, ports, docks, log dumping, ocean dumping. Land-based impacts include industry, pulp and paper, onshore mining, human settlements, forestry, and agriculture. Climate change impacts include ocean acidification, sea level rise, sea temperature change, and UV change. When all drivers were categorized, we calculated the proportion of direct versus indirect impacts (also accounting for impacts that could be both or neither) within each category affect each ecosystem service.

## 2.6 Cumulative impacts model

After all ecosystem services were modeled, their spatial overlap with all activity and stressors was mapped at a 500x500m cell resolution. The spatial extent of specific ecosystem services served as the boundary for each overlapped map. All intensity data for drivers were log transformed and normalized by dividing by the largest intensity value found for each driver across the BC coast to generate a dimensionless 0–1 intensity scale [33]. Cumulative impact $I_c$ was calculated for each pixel according to the established cumulative impact map formula

$$I_c = \sum_{i=1}^{n} D_i \times E_j \times \mu_{i,j}$$

where $D_i$ is the log-transformed and normalized intensity scores for driver $i$, $E_j$ is the presence or absence of ecosystem service $j$, and $\mu_{i,j}$ is the risk of individual occurrences of driver $i$ on ecosystem service $j$ (see Supplementary Methods, 33). Cumulative impacts were calculated twice: first, cumulative impact scores were calculated without the service and value dimensions (i.e. only considering ecosystem service supply); next, cumulative impact scores were calculated with service and value dimensions. The difference between these two calculations reveals the contribution of considering the service and value dimensions when assessing cumulative impacts on ecosystem services. These cumulative impact scores were calculated both across the spatial range of each ecosystem service as well as calculated per-cell (at a 500mx500m cell resolution).

## 3. Results

### 3.1 Impacts to ecosystem services

**3.1.1 Per cell cumulative impacts to ecosystem service supply, service, and value.** Our results indicated that all modeled ecosystem services are impacted across most–if not all–of their range (Figs 2 and 3). Controlling for total range, benefits from commercial demersal fisheries were ranked highest for average per-cell cumulative impact ($I_c$) from drivers, followed by benefits from commercial pelagic fisheries, potential renewable energy, coastal recreation, benefits from finfish aquaculture, benefits from shellfish aquaculture, coastal protection, and aesthetics (Fig 4 and Table 2).

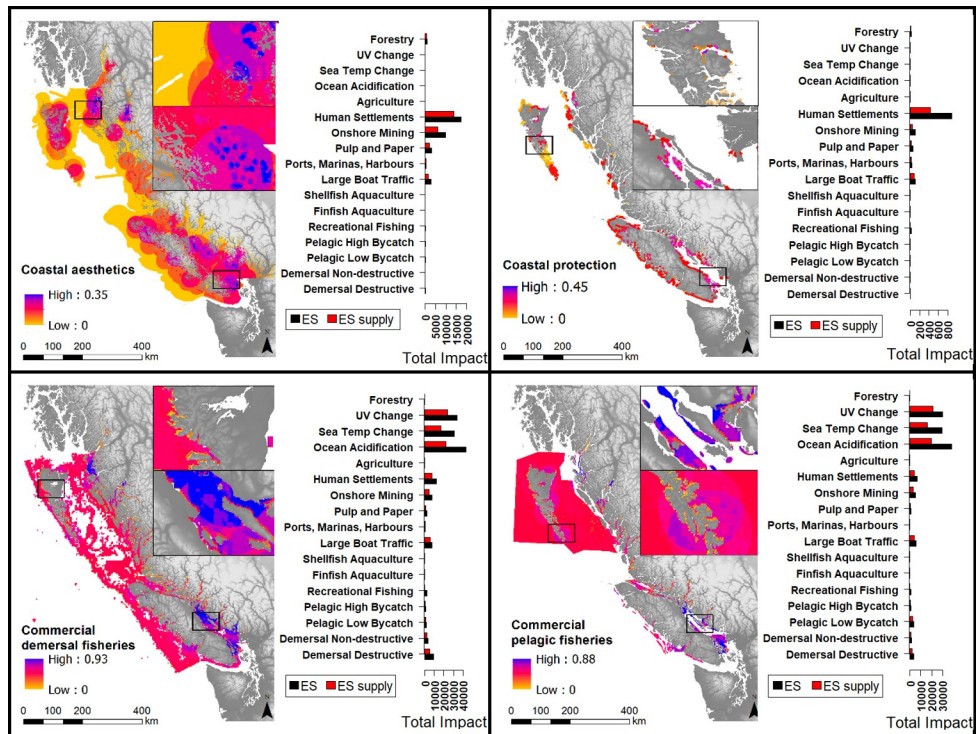

**Fig 2. Cumulative impact maps for four ecosystem services (aesthetics, coastal protection, benefits from commercial demersal fisheries and benefits from commercial pelagic fisheries), with associated bar graphs of drivers of impact.** Maps display the summed impact of all drivers to each ecosystem service; bar graphs show total impact values for each driver. Red bars indicate impact only accounting for ecosystem service supply dimensions (ES supply), and black bars indicate impact accounting for the entire ecosystem service cascade, including supply, service, and value (ES). Coastal protection is not to scale to allow for visibility. Four drivers that cause impact have been left off the bar graphs because they contribute negligible levels of impact across ecosystem services (small docks, log dumping, ocean dumping, and industry).

**3.1.2 Per cell cumulative impacts to ecosystem service supply.** When mapping the per-cell cumulative impact model while only considering impacts to ecosystem service supply (and not including service and value dimensions), the ranked list of ecosystem services facing impacts is similar to the list considering service and value dimensions, with some differences. Benefits from commercial demersal fisheries are still ranked highest, followed by benefits from shellfish aquaculture, benefits from commercial pelagic fisheries, benefits from finfish aquaculture, coastal recreation, potential renewable energy, coastal protection, and coastal aesthetics (Table 2). However, all ecosystem services vary greatly in their relative impacts (Fig 4). Most ecosystem services have per-cell $I_c$ values that range from ~0–0.8, except aesthetics, which only has $I_c$ values ~0–0.4 (Fig 3).

**3.1.3 Cumulative impacts to ecosystem service supply, service and value across spatial range.** The total cumulative impact scores ($I_c$) across the spatial range of ecosystem services were highest for benefits from commercial demersal fisheries, followed by benefits from commercial pelagic fisheries, aesthetics, coastal recreation, potential renewable energy generation, coastal protection, benefits from finfish aquaculture and benefits from shellfish aquaculture (Figs 2 and 3, and Table 2). For many ecosystem services, higher levels of impact were found on the south of the coast, between Vancouver Island and the mainland (for benefits from finfish and shellfish aquaculture and potential energy generation, and coastal protection), and the north coast (for aesthetics, coastal protection, demersal and pelagic fisheries, and marine

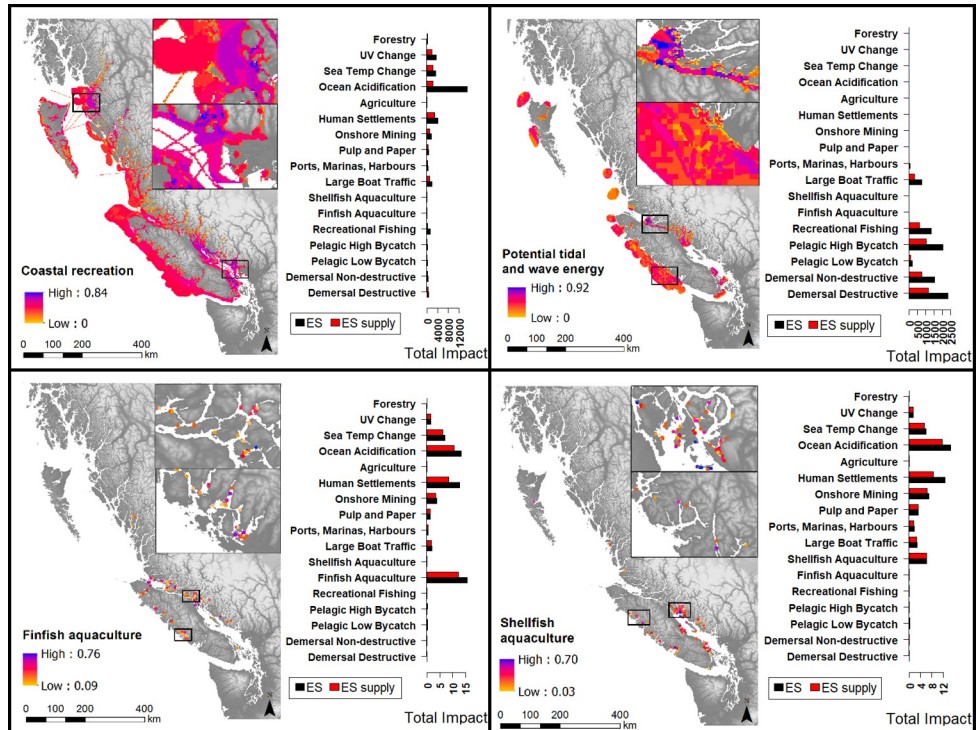

**Fig 3. Cumulative impact maps for four ecosystem services (recreation, energy, benefits from finfish aquaculture, and benefits from shellfish aquaculture), with associated bar graphs of drivers of impact.** Maps display the summed impact of all drivers to each ecosystem service; bar graphs show total impact values for each driver. Red bars indicate impact only accounting for ecosystem service supply dimensions (ES supply), and black bars indicate impact accounting for the entire ecosystem service cascade, including supply, service, and value (ES). Aquaculture sites are not to scale to allow for visibility. Four drivers that cause impact have been left off the bar graphs because they contribute negligible levels of impact across ecosystem services (small docks, log dumping, ocean dumping, and industry).

recreation, Figs 2 and 3). Major hotspots of impact are similar when considering service and value dimensions of impact versus not considering them.

**3.1.4 Cumulative impacts to ecosystem service supply across spatial range.** Considering only ecosystem service supply dimensions in calculating cumulative impact ($I_c$), the ranked list of ecosystem services facing the most severe impact is largely consistent with the ranking of when service and value dimensions are also considered; however, the position of benefits from finfish aquaculture and benefits from shellfish aquaculture are switched (Figs 2 and 3, and Table 2).

**3.1.5 Prominent drivers of impact.** Different groups of drivers generated prominent impacts for different ecosystem services (Figs 2 and 3). Climate related stressors contributed high levels of impact to benefits from demersal and pelagic fisheries, marine recreation, benefits from finfish aquaculture and benefits from shellfish aquaculture. Ocean acidification was the main climate related stressor contributing to impact in these ecosystem services. Climate related stressors had the highest spatial range across all ecosystem services (occupying all map cells). Land-based activities contributed high levels of impact to aesthetics, coastal protection, and both aquaculture categories. Human settlements and onshore mining contributed the most impact to most of these ecosystem services. Coastal commercial activities contributed high levels of impact to benefits from finfish aquaculture. Aquaculture was seen as a prominent activity impacting itself, as experts scored risk to ecosystem service supply, service, and value dimensions high for aquaculture, and multiple experts described the self-harmful practices

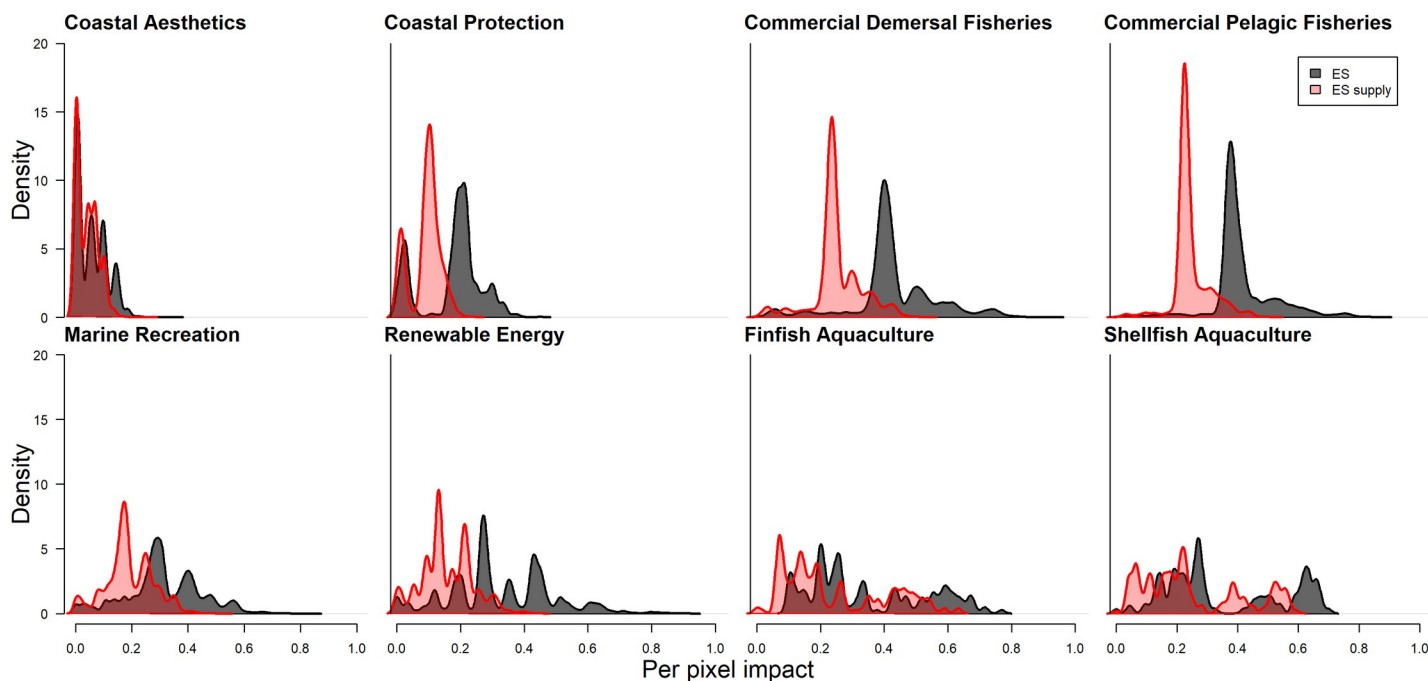

**Fig 4. Density histograms of per-cell $I_c$ values for each ecosystem service.** Red histograms indicate per-cell impact only accounting for ecosystem service supply (ES supply), and black histograms indicate impact accounting for all dimensions, including supply, service, and value (ES).

and invasive and disease problems of aquaculture. They also cited the poor public attitude towards aquaculture as a high risk to itself. Fisheries contributed high levels of impact to potential tidal and wave energy. Experts scored risk to service and value dimensions from fisheries to potential energy generation high, specifically the effects of fisheries on access to good renewable energy sites.

**Table 2. Per-cell and total cumulative impact scores for all ecosystem services.** Cumulative impact scores are provided for the models considering impact across ecosystem service cascade (supply, service, and value), as well as for the models considering impact only to ecosystem service supply.

| Ecosystem Service | Per-cell cumulative impact (per-cell $I_c$) for ecosystem service supply, service, and value | Per-cell cumulative impact (per-cell $I_c$) for ecosystem service supply | Total cumulative impact ($I_c$ across spatial range) for ecosystem service supply, service, and value | Total cumulative impact ($I_c$ across spatial range) for ecosystem service supply |
|---|---|---|---|---|
| Coastal Aesthetics | 0.058 | 0.040 | 35393.548 | 24854.244 |
| Coastal Protection | 0.178 | 0.090 | 798.622 | 404.969 |
| Benefits from Commercial Demersal Fisheries | 0.431 | 0.253 | 156985.599 | 92249.122 |
| Benefits from Commercial Pelagic Fisheries | 0.416 | 0.246 | 127730.680 | 75563.987 |
| Coastal Recreation | 0.312 | 0.189 | 21067.009 | 12825.409 |
| Potential Energy Generation | 0.325 | 0.157 | 8264.231 | 4000.945 |
| Benefits from Finfish Aquaculture | 0.303 | 0.243 | 58.162 | 46.749 |
| Benefits from Shellfish Aquaculture | 0.289 | 0.249 | 54.436 | 46.941 |

### 3.2 Importance of service and value metrics to impact scores

Across all ecosystem services, total and per-cell impact scores were more severe when including risk to service and value dimensions in impact calculations than excluding them (Figs 2, 3 and 4). Resulting maps show greater overall impact across the spatial range of all ecosystem services when these service and value dimensions are included on top of ecosystem service supply dimensions (S1 Fig). Though we use an additive model (and so any additional criteria will add to total impact), the service and value dimensions contributed a substantial proportion towards total impact (Fig 4). Including these service and value dimensions had the greatest proportional increase in per-cell $I_c$ for potential renewable energy generation, followed by coastal protection, benefits from commercial demersal fisheries, benefits from commercial pelagic fisheries, recreation, aesthetics, benefits from finfish aquaculture, and benefits from shellfish aquaculture (Table 3 and Fig 4). Considering total $I_c$ values (across the spatial range) including service and value dimensions had the same proportional increases in cumulative impact scores (Table 3). The only case where considering impacts on service and value dimensions did not add to impact estimates was the impact of shellfish aquaculture on itself (Fig 3).

### 3.3 Future risk to ecosystem services

Considering future risk, experts perceived that some ecosystem services are at greater risk from some future climate stressors than potential major oil spill, while others are at greater risk from potential major oil spills (Fig 5). Aesthetics, coastal protection, and potential energy generation were all perceived to be at higher risk from a major oil spill on the coast, and face no risk from future sea temperature or ocean acidification. Coastal protection and potential energy generation were perceived to be at high risk from sea level rise, but we did not have spatial data for this stressor so we do not represent it here. In contrast, benefits from fisheries, benefits from aquaculture, and marine recreation all appeared to be at higher risk from future ocean acidification and sea surface temperature rise, and particularly ocean acidification.

### 3.4 Relative importance of ecosystem service supply, service, and value for impact

Based on expert ranking, risk to ecosystem services is dependent on diverse criteria of exposure and consequence, without a clearly dominant criteria influencing risk (Fig 6). For

**Table 3. The relative (proportional) increase in per-cell and total cumulative impact scores when modeling impacts to ecosystem service supply, service, and value for each ecosystem service compared to only modeling impacts to ecosystem service supply.**

| Ecosystem Service | Proportional increase in per-cell cumulative impact (per cell $I_c$) and total cumulative impact ($I_c$ across spatial range) from considering impact to ecosystem service supply, service, value compared to only ecosystem service supply |
|---|---|
| Coastal Aesthetics | 0.42 |
| Coastal Protection | 0.97 |
| Benefits from Commercial Demersal Fisheries | 0.7 |
| Benefits from Commercial Pelagic Fisheries | 0.69 |
| Coastal Recreation | 0.64 |
| Potential Energy Generation | 1.07 |
| Benefits from Finfish Aquaculture | 0.24 |
| Benefits from Shellfish Aquaculture | 0.16 |

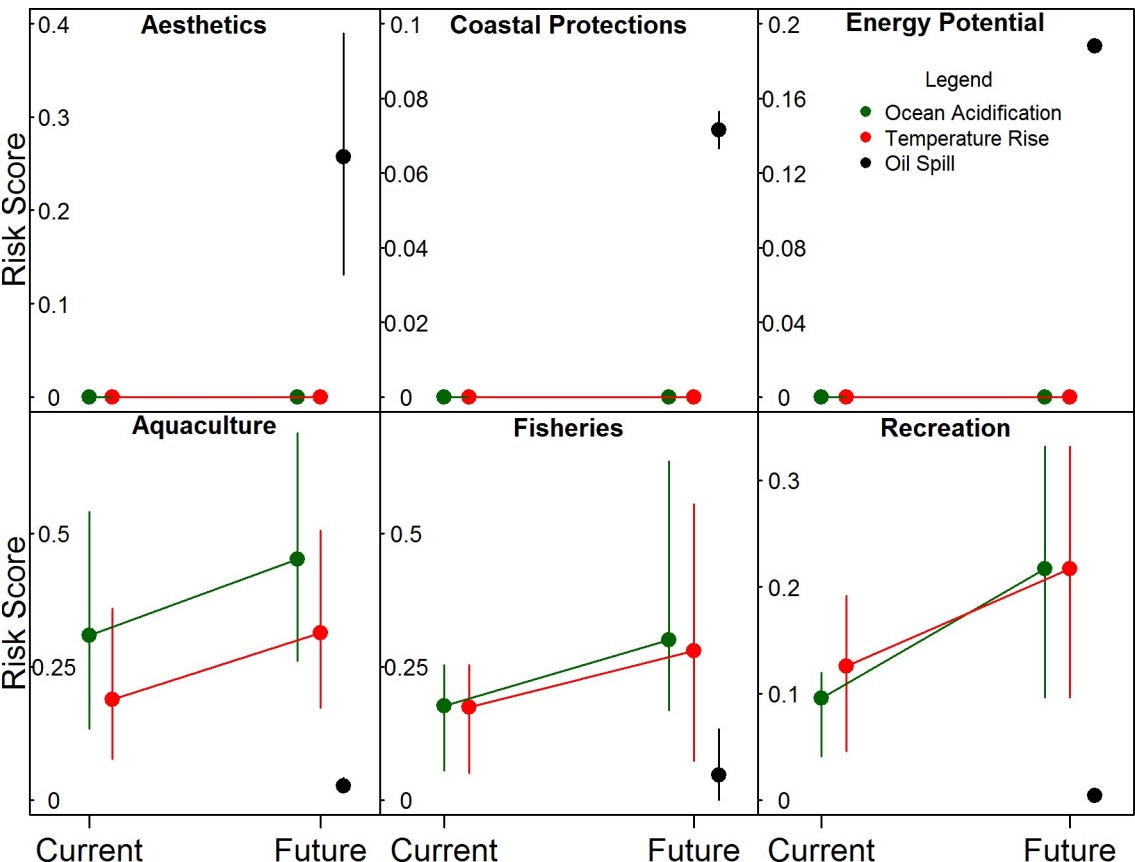

**Fig 5. The risk posed by future climate change risks and oil spills on six ecosystem services, compared with current climate change risks.** Points represent mean risk scores, error bars represent 25th and 75th percentiles, and lines connecting points demonstrate the trajectory of risk from current conditions to future conditions.

exposure criteria, experts considered the spatial extent of individual occurrence of activities to be most important, followed by the recovery time of an ecosystem service to an impact, and finally the frequency at which an ecosystem experiences an activity. For consequence criteria, experts considered the magnitude of change to the biophysical processes that produce the ecosystem service to be most important, followed by how the perceived quality of an ecosystem

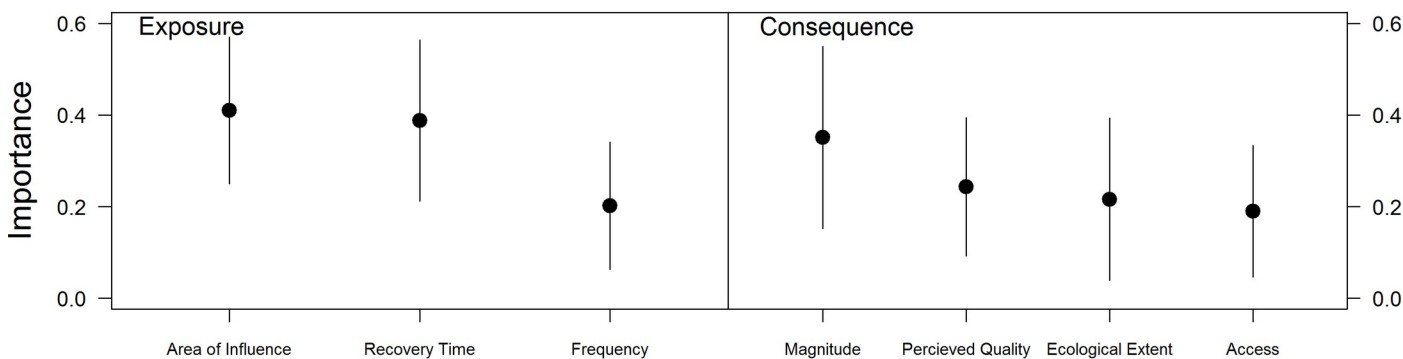

**Fig 6. The perceived importance of risk criteria to exposure and consequence.** Points and error bars represent mean and standard deviations of the distribution of relative importance of risk criteria.

service changes in response to an impacting activity, the extent to which the environment is impacted (from individual species to entire ecosystems), and finally the changes to access to an ecosystem service. However, simple rankings mask the finding that experts perceived all criteria to contribute non-trivially to risk (the best model estimated frequency to contribute 20% to exposure, and access to contribute 19% to consequence), and that there was a diversity of weights considered across our experts (Fig 6), reflecting that some experts considered service and value dimensions of ecosystem services to be more important than biophysical supply components.

### 3.5 Understanding mechanisms of impact

Experts suggested diverse prominent mechanisms of effect from drivers of impact among the ecosystem services (Fig 7). Across all types of impact, including fisheries impacts, coastal commercial activities, land based activities and climate stressors, some ecosystem services have consistent impact mechanism types. Most aesthetics experts suggested that impact mechanisms to aesthetics are direct, with some specifically suggesting that the physical footprint of the activity is often all that matters for aesthetics. Renewable energy potential was an ecosystem service that many experts suggested was not affected by any driver, though a sizeable minority suggested that fisheries affect it directly through restricting access, and that climate change affects it both directly and indirectly through changing sea levels and affecting energy demand (which affects the infrastructural needs and suitability of locations for energy sites). Coastal protection was most often thought to be directly affected by drivers through physical damage to kelp and seagrass beds and through pollution, and some suggested that recreational fishing vessels crowd estuaries and fjords, destroying habitat that support wave attenuation, and themselves generate additional wake that can risk coastlines. Most experts suggested that benefits

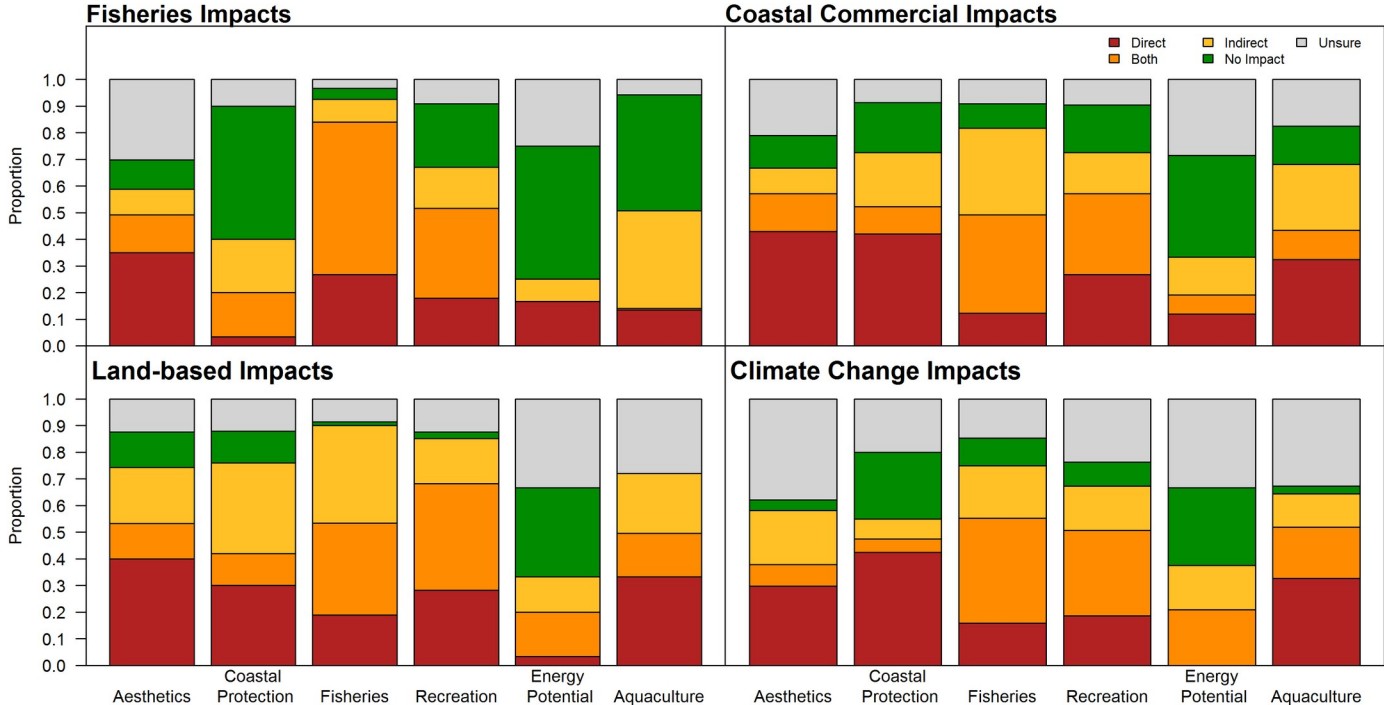

**Fig 7. The proportion of each type of impact mechanism (direct, both direct and indirect, indirect, no impact, and unsure) from four categories of drivers to the eight ecosystem services, as indicated by experts.**

from aquaculture are predominantly directly affected by some drivers (such as land based run-off) but indirectly through others (such as invasive and disease spread from fishing vessels and ships), as well as directly and indirectly from sea temperature and ocean acidification affecting the harvested species as well as organisms that they feed on. Fisheries and recreation were both suggested to face both direct and indirect impacts from climate change, fisheries, coastal commercial, and land-based sources, according to experts. Many experts suggested that changes to foodwebs and other ecological dynamics result in indirect impacts along with direct impacts from all types of human impacts.

## 4. Discussion

### 4.1 Including service and value dimensions lead to greater accounting of impact

Considering service and value dimensions in addition to ecosystem service supply leads to more severe cumulative impact scores and a greater diversity of impact pathways. However, our results indicate that consideration of service and value dimensions does not greatly affect a relative understanding of impact across ecosystem services: only considering ecosystem service supply generated similar ranks of ecosystem services facing greatest impact and highlighting hotspots of impact (though we found some differences in ranking of ecosystem services when assessing per-cell impact). Our results may be interpreted to suggest that impact maps of ecosystem services that only consider supply dimensions may accurately generate conclusions about what services face greatest impact and where they face greatest impact. However, considering service and value dimensions set the scope of which services are considered for impact assessment (by determining which services are most valued) and their spatial boundaries (because people do not benefit from ecosystem services throughout their entire range of biophysical production). Additionally, ours is an initial investigation into the importance of service and value dimensions for ecosystem service impact, and expert scores of risk criteria may fail to emphasize service and value dimensions because of two important biases. First, many of the experts taking part in our survey have ecological and biophysical training. Second, most prominent frameworks of ecosystem service change represent impacts as mediated solely through the biophysical community [23,25,26], which may affect how experts think about impacts. In cases where there are important impacts that overwhelmingly impact ecosystem services through service and value dimensions, excluding these dimensions may lead to different rankings of threatened ecosystem services and different map hotspots. Determining how prevalent these cases are in different settings remains to be seen. Regardless of understanding relative impact, our results indicate that studies based only on supply dimensions may under-represent the processes that generate impact to ecosystem services. Considering the ecosystem service cascade from service supply through service delivery through satisfying values [13] may lead to a more detailed understanding about impacts and potential responses to these impacts.

### 4.2 Mapping ecosystem services allows for insights not afforded by mapping habitats

Impact on ecosystem services is a function not only of spatial overlap with concurrent activities and stressors, but of the risk through ecosystem service supply, service, and value dimensions of ecosystem services as well. Many ecosystem services face high impact in the area between Vancouver Island and the mainland, a finding reinforced by previous studies that focused solely on impacts to habitats [4,8,10,31,40]. Not all ecosystem services have impact hotspots

here, however, reflecting the importance of accurately mapping ecosystem services. While our study alongside previous ones may share similar patterns of human activity, the distribution of ecosystem services themselves is important in determining where areas of high impact are. The marine InVEST models use data of environmental process and human activity to spatially represent ecosystem services, allowing us to directly model ecosystem services [21].

Accurately representing the overlap of activities and stressors on ecosystem services generate additional insights. Knowing where ecosystem services are at highest risk can allow managers to assess impact relative to areas of high demand [28]. Areas of high risk to coastal protection were concentrated close to population centers (in the southern Strait of Georgia), partly because the human activities that might benefit the most from coastal protection–human settlements–also provide the largest impact to coastal protection. Spatial representation also allows for an understanding of whether an ecosystem service faces high risk on account of large spatial range despite low per-area impact (such as aesthetics), versus ecosystem services that face low total impact because of limited geographic range despite having high per area impact (such as benefits from shellfish and finfish aquaculture). Aesthetics was found to be the least impacted ecosystem service per unit area, indicating that a beautiful coast may mask a highly impacted coast.

Explicit inclusion of risk criteria (exposure and consequence) is important because activities and stressors with extensive spatial range and high overlap with ecosystem services do not necessarily generate high impact. Similar to a recent cumulative impact mapping study on coastal ecosystems in British Columbia [10], and along the California current [4] we found climate change impacts to be important stressors (especially ocean acidification), highlighting the importance of their inclusion in analysis, and cautioning results from mapping studies that do not include them [8]. For example, ecosystem services dependent on invertebrate and finfish (benefits from fisheries, benefits from aquaculture, and marine recreation) were highly impacted by ocean acidification. Climate change drivers exist across the entire marine system along the British Columbia coast and consequently fully overlap with every ecosystem service we modeled, yet impact on some ecosystem services is driven largely by climate impacts (such as benefits from fisheries and benefits from aquaculture) while others are largely indifferent to climate change stressors (such as aesthetics, coastal protection and potential energy generation). Indeed, kelp and seagrasses associated with coastal protection may benefit with ocean acidification [41,42]. Sea-level rise was indicated as a high risk stressor especially to coastal protection, but we did not have spatial data for sea level rise. While the risk scores for potential energy generation are uncertain given the input from only one expert, the conclusion that energy generating infrastructure and planning faces greater risk from human activities than climate impacts is plausible.

This dichotomy between global and regional impacts may exacerbate in the future, as experts suggested that future climate change impacts (specifically warming and ocean acidification) will be a higher risk to those climate-sensitive ecosystem services compared to current conditions, while climate-insensitive ecosystem services will face similar risk levels. For these latter ecosystem services, potential future development may pose a greater cause for concern. Future oil spill potential related to planned developments of oil and gas with associated marine shipping poses a significant risk to these ecosystem services. Previous efforts to compare climate change impacts with future developments in British Columbia indicated that climate change has greater regional scale impact across ecosystem types but lower local impact [10]. We show that some ecosystem services–in contrast to ecosystem types–show varying degrees of risk to different types of stressors, leading to insensitivity to climate change stressors for some ecosystem services at local and regional scales.

## 4.3 Service and value dimensions are important for understanding causes of impact

Experts in our survey treated individual service and value dimensions with comparable importance to supply dimensions when ranking scenarios. Service and value dimensions are definitional to ecosystem services yet are often overlooked in quantitative assessments. Despite supply dimensions potentially being sufficient for understanding which ecosystem services are most impacted relative to one another, relying on supply dimensions alone is shortsighted for two reasons. First, any quantitative measure of impact is likely to be an underestimate [14]. Service and value factors, such as how people perceive an ecosystem service, can regulate the extent to which people enjoy and benefit from the ecosystem service [27]. For example, open-pen finfish aquaculture practices are perceived negatively by many people in British Columbia [29], creating a self-stigmatized industry. Whether public perceptions on finfish aquaculture are warranted or not, they affect aquaculture as the aquaculture industry has launched marketing campaigns to fight its reputation (www.bcsalmonfacts.ca). Second, many ecosystem services can be impacted largely (even solely) through changes to access and perceived value. Experts indicated that potential wave and tidal energy production face risk from fisheries and ports partly through the competition for space, as access to suitable power generation sites can be blocked or zoned out by competing interests for the area. If situations where impacts occur through service and value dimensions become more common then relying on supply dimensions may no longer be suitable for understanding which ecosystem services face highest relative impact.

## 4.4 Accounting for pathways of impact can improve cumulative impact models

Ecosystem services may require different data and representation techniques than ecosystem types. Unlike ecosystem types, ecosystem services are not variants of geographical classes; ecosystem services do not only exist on a landscape but are related to people's values and ability to obtain them [27,43]. The same activity may have different impact pathways on two different ecosystem services because one ecosystem service could be primarily impacted through a change in species density while another could be impacted primarily because the activity restricts people to a region through property rights and trespassing laws. The greater diversity of potential pathways of impact that ecosystem services face arguably puts greater emphasis on understanding the causal processes of impact for ecosystem services than for ecosystem types.

Given the diverse kinds of ecosystem services that exist, a common spatial representation of specific human activities and stressors across ecosystem services may produce misleading results in two important ways. First, the impact pathway important for the ecosystem service should dictate the size of the zone of influence [8]. Many experts in our study suggested that aesthetics are directly impacted from most activities and stressors, and that what matters is the physical footprint of any activity. We have onshore mining spatially represented to account for acid mine drainage and tailings that occur kilometers away from mines themselves. This area of influence is likely appropriate when mapping impact to ecosystem services affected by these processes, such as benefits from fisheries and aquaculture, but it may lead to overestimated overlap of mining impacts and aesthetics. Future efforts to map impacts on ecosystem services should match the spatial representation of activities with relevant impacts. Second, not all experts understand the impact pathways the same, which means they may not answer the same questions. The precedent set here using expert surveys, in conjunction with impact mapping, asks experts to assess vulnerability/risk to an activity "considering all relevant impacts". This open question framing allows for a tractable survey, yet our results suggest that what is considered in "all relevant impacts" may vary from expert to expert for a given human activity.

What's hidden in our resulting maps is a significant epistemic uncertainty that can be reduced with appropriate elicitation strategies [44]. Future expert elicitation processes should emphasize specific pathways when assessing risk, even if it means batching surveys into sets of different impact pathways so different experts quantify risk to different impacts.

## 4.5 Limitations and opportunities

While we present advancement in cumulative impact mapping–namely representing ecosystem services and accounting for impact along supply, service, and value dimensions–and recommend data considerations specifically for ecosystem services, we must also acknowledge persistent limitations of impact mapping. Most importantly these include a static representation of impact and a simplistic model of cumulative impacts [3,8,31]. Though experts considered temporal criteria of exposure as less important than area of influence, they were still important components of risk, showing that temporal considerations are essential. Spatial models are often snapshots in time, and though we include some temporal dynamics (assessing risk of foreseen impacts) there are many important temporal aspects of impact that are not captured. We do not represent future impacts spatially (but see Murray et al. [10] for a spatial analysis of proposed projects), though understanding future impacts would be highly valuable to managers. We also do not account for historic impacts. By focusing on contemporary impacts we set a contemporary benchmark and could not consider change from ecosystem service states that may be more ideal, such as times in the past when overfishing was not as prevalent [45,46]. As a simple model of cumulative impacts we also could not explicitly represent some important ecological dynamics and cascading effects that do not co-occur spatially. For example, bottom trawling can negatively affect nursery grounds for species that are fished in other locations. While we tried to capture some of these dynamics by asking experts to score the "community extent" of risk, we cannot capture the full complexity of human impact on ecosystem function, particularly where impacts affect habitats and ecosystem functions underpinning services that occur elsewhere.

The cumulative impact model we employ assumes an additive, relative model of impact with no upper bound. Both activity intensity and risk scores were normalized between 0–1, so components of the model have measurement boundaries, but the cumulative impact can aggregate indefinitely. Empirical studies have shown additive cumulative impacts to occur in a minority of situations [47,48]. Synergistic impacts–when the total impact is greater than the sum of component impacts–occur often, especially when more than two impacts co-occur [5,47,49]. Antagonistic impact–when total impact is less than the sum of component impacts–are also prevalent, and have been shown when global impacts interact with local impacts [50,51]. The theoretically limitless measure of impact produced by the model employed here also assumes that impacts can accumulate indefinitely, and that thresholds do not exist [3]. These are obviously false assumptions, but this model can still provide broad insights into the relative impact faced by multiple ecosystem services.

Finally, this work depends on input from experts. Expert input can be affected by biases [34] and therefore can increase uncertainty of results. Uncertainty in expert responses is even higher when few experts provide input (such as for potential wave and tidal energy generation here), and responses should be considered as hypotheses requiring empirical validation [6]. However, where empirical data does not exist (such as in the case here), expert input elicited through structured processes can provide valuable input for decision-making [25,30,34], including the specific elicitation techniques used here [29].

Despite modeling limitations, mapping cumulative impacts to ecosystem services allows for unique planning opportunities. Ocean managers can use this approach to explore the spatial

feasibility of potential coastal uses, as we show for potential wave and tidal power generation, even if precise risk estimates are not available. By mapping areas of potential energy generation, we see that the areas of lowest threat to energy generation are the central coast and some areas between Vancouver Island and the mainland. These are relatively unpopulated areas, which may mean higher infrastructure costs to establish turbines, but these costs may be worth avoiding impediments in more populous areas.

## 5. Conclusion

By mapping cumulative impacts to ecosystem services, we can better steward our ecosystems and understand the dual relationship of humans to the environment: as agents of change and beneficiaries of services [12]. We have demonstrated the kinds of rich insights that can be gained from mapping impacts to ecosystem services, including: 1) discovering where, and by what means, different ecosystem services face the greatest impact; 2) determining what ecosystem services are comparatively worse (or better) off under current conditions; 3) understanding the ways in which impacts manifest; 4) assessing spatial feasibility for new ocean uses. We have also demonstrated the importance of considering service and value dimensions in assessing impact. We argue that considering service and value dimensions is not only important to more fully understand impact, but also to plan effective management responses. Finally, we have pointed to areas of future methodological refinement, and encourage greater innovation in cumulative impact mapping. Ecosystem services can be highly location specific [27], so future risk assessments are warranted in new places. Understanding risk and impact to ecosystem services should be an essential management priority to maintain the flow of services we benefit from.

## Supporting information

**S1 Table. Erosion risk to different coastal classes.**
(DOCX)

**S2 Table. Data files, sources, and resolution used to map impacts and ecosystem services.**
(DOCX)

**S3 Table. Descriptions of human activities and stressors provided to experts to assess risk.** Note that some category descriptions here describe multiple human activity and stressor data layers (from S2 Table). They are grouped to avoid repetition.
(DOCX)

**S4 Table. Descriptions of exposure criteria given to experts to assess risk.**
(DOCX)

**S5 Table. Descriptions of consequence criteria given to experts to assess risk.**
(DOCX)

**S6 Table. Normalized expert scores for the seven risk criteria of different human activities for the ecosystem services.** Values represent means and standard errors in brackets.
(DOCX)

**S1 Fig. Side by side comparison of impact maps considering all risk criteria, including ecosystem service supply, service, and value (maps on the left) versus only considering biophysical criteria of risk which only assesses impact to ecosystem service supply (maps on the right).** Map pairs are for A) aesthetics, B) coastal protection, C) benefits from commercial demersal fisheries, D) benefits from commercial pelagic fisheries, E) coastal recreation, F)

potential renewable energy, G) benefits from finfish aquaculture, and H) benefits from shellfish aquaculture.
(DOCX)

**S1 File.**
(DOCX)

**S2 File. Copy of the expert survey.**
(DOC)

## Acknowledgments

We are grateful to A. Thompson and T. Coyle for expert identification and help building the online survey. We would also like to thank G. Peterson, M. O'Connor, and S. Gergel for reviewing and commenting on the work. We would also like to thank WWF Canada for the use of data on human activities.

## Author Contributions

**Conceptualization:** Gerald G. Singh, Benjamin S. Halpern, Terre Satterfield, Kai M. A. Chan.

**Data curation:** Gerald G. Singh.

**Formal analysis:** Gerald G. Singh, Ian M. S. Eddy, Rabin Neslo.

**Funding acquisition:** Gerald G. Singh, Kai M. A. Chan.

**Investigation:** Gerald G. Singh.

**Methodology:** Gerald G. Singh, Ian M. S. Eddy, Benjamin S. Halpern, Rabin Neslo, Terre Satterfield, Kai M. A. Chan.

**Project administration:** Gerald G. Singh.

**Resources:** Gerald G. Singh.

**Software:** Gerald G. Singh, Ian M. S. Eddy.

**Supervision:** Gerald G. Singh, Benjamin S. Halpern, Terre Satterfield, Kai M. A. Chan.

**Validation:** Gerald G. Singh.

**Visualization:** Gerald G. Singh, Ian M. S. Eddy.

**Writing – original draft:** Gerald G. Singh.

**Writing – review & editing:** Gerald G. Singh, Ian M. S. Eddy, Benjamin S. Halpern, Rabin Neslo, Terre Satterfield, Kai M. A. Chan.

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
