## [Decision Letter · Decision Letter 0]

4 Aug 2019

PONE-D-19-18924

Mapping Cumulative Impacts to Coastal Ecosystem Services in British Columbia

PLOS ONE

Dear Dr. Singh,

Thank you for submitting your manuscript to PLOS ONE. After careful consideration, we feel that it has merit but does not fully meet PLOS ONE’s publication criteria as it currently stands. Therefore, we invite you to submit a revised version of the manuscript that addresses the points raised during the review process.

Both text and figures have to be reorganized according to reviewers' comments.

We would appreciate receiving your revised manuscript by Sep 18 2019 11:59PM. To enhance the reproducibility of your results, we recommend that if applicable you deposit your laboratory protocols in protocols.io, where a protocol can be assigned its own identifier (DOI) such that it can be cited independently in the future. For instructions see: http://journals.plos.org/plosone/s/submission-guidelines#loc-laboratory-protocols

We look forward to receiving your revised manuscript.

Kind regards,

Carlo Nike Bianchi

Academic Editor

PLOS ONE

Journal Requirements:

1. Please include additional information regarding the survey or questionnaire used in the study and ensure that you have provided sufficient details that others could replicate the analyses. For instance, if you developed a questionnaire as part of this study and it is not under a copyright more restrictive than CC-BY, please include a copy as Supporting Information.

Reviewers' comments:

Reviewer's Responses to Questions

**Comments to the Author**

1. Is the manuscript technically sound, and do the data support the conclusions?

Reviewer #1: Yes

Reviewer #2: No

2. Has the statistical analysis been performed appropriately and rigorously? 

Reviewer #1: Yes

Reviewer #2: No

3. Have the authors made all data underlying the findings in their manuscript fully available?

Reviewer #1: No

Reviewer #2: No

4. Is the manuscript presented in an intelligible fashion and written in standard English?

Reviewer #1: Yes

Reviewer #2: No

5. Review Comments to the Author

Reviewer #1: The study presented here provides a novel method for estimating the human activities impact on ecosystem services. This is a very interesting and important field of research, among others because it provides valuable knowledge that could be applied to informed management and decision making. From the scientific point of view, the linkages between human activities, the pressures they produce and the associated environmental impact and their consequences in ecosystem services provisioning, is still a very complex topic and not well-known.

Adopting the British Columbia as case study, the authors describe a comprehensive process conducted for the obtention of the input data, model development and implementation. In one hand, the authors have estimated the ecosystem services production and their spatial distribution by using real data or using already available and broadly used models (InVEST) models for different ecosystem services). On the other hand, authors have used information dealing with the distribution of human activities. The most challenging point of the process is the estimation of the impact of each activity on each of the ecosystem services analysed. The authors have solved this by performing a consultation to experts. Not having empirical data that link human activities to their impacts, authors have adopted this method for obtaining the information that afterwards was needed to develop the models. All the methodology is exhaustively described in the Supplementary Material.

The methodology and the results presented could be deeply discussed and critised, but the authors are aware of the limitations and the need of adoption of assumptions during the implementation of the approach. For this reason, the authors dedicate a full subsection of the paper (Section 4.5), acknowledging the weaknesses and limitations of the approach.

Specific comments

Revise and complete the affiliations.

I miss some more information regarding to the study site to justify why this area was selected for the research. How is been managed at present? Additional information such as the coastal length, surface or area of the case study, depth ranges. A very short description. A couple of lines would be enough.

Line 166. “Bathymetry and topography was used to calculate viewshed”. I would say that what is below the sea surface can not be seen and that is not part of the viewshed.

Lines 272-274, would need a rephrasing to make it more clear.

Section 4.5. As stated previously, this is an important section of the paper because the authors list all the limitations and assumptions that they had to adopt during the modelling process. Nevertheless, I still miss one more point. The authors are assuming that the ecosystem services production and the location of the maritime activity overlap. This is an assumption needed when operating between different information layers, but I think that this should be also discussed. Certain activities could have an effect on ecosystem services that are produced elsewhere. For example, fishing activity (e.g. bottom trawling), could impact nursery grounds of species of commercial interest that are fished in other locations. Thus, the impact on the ecosystem service and the activity are not spatially coincident. The same happens for other ecosystem services. This is a complex point of ecosystem functioning but that it would be interesting to mention in this section.

Figure 1. I think that it would be interesting to add labels to the axis of the bar charts.

Figure 2. It would be interesting to add a graphical legend indicating what are representing the red and grey graphs.

Figure 3. Add axis labels.

I have added some extra comments in the Supplementary Material.

Reviewer #2: The aim of the paper is very good and the subject is novel, very interesting and very topical. Nonetheless results are presented in a confused way: the main problem in my opinion resided in the fact taht authors wanted to convey to much information. Some results have then been reported in supplementary materials while several results paragraphs deals with the same figures. As a consequence the reader is not allowed to a full understanding.

I suggest to reorganise the paper and

1. to be consistent and to use the same terms in methods and results

2. it would be useful to have the same sections in methods and results and in the same order of appearance

3. choose most important results and information and focus on them better explaining how they are obtained

I made comments until the results section, for discussion i would like to see a revised verison of the paper

Here below my specific comments: please pay particular attention to comments related to the intro section

Introduction

I appreciate that authors cite the ecosystem services cascade that represents, in my opinion, a turning point concerning the ecosystem services theory. Nonetheless I am not sure to share their interpretation about the cascade.

The cascade framework was developed in order to highlight the dependence of human well-being to ecosystems. As the authors (Haines-Young, R., & Potschin, M. (2010). The links between biodiversity, ecosystem services and human well-being. Ecosystem Ecology: a new synthesis, 1, 110-139.) of the cascade say “Ecologists will increasingly have to work alongside economists, geographers and a range of other social scientists to understand the value that biodiversity and ecosystem services have, to assess the costs and benefi ts of different conservation and management strategies, and to help design the new governance systems needed for sustainable development. Biodiversity has intrinsic value and should be conserved in its own right. However, the utilitarian arguments which can be made around the concept of ecosystem services and human well-being are likely to become an increasingly central focus of future debates about the need to preserve ‘natural capital’. The wider research community needs to engage in such debates. Although long-term sustainable development has come to mean many things, the concept must include the maintenance of ecosystem services and the elements of human well-being that depend upon healthy ecosystems. If the Ecosystem Approach is to be embedded in decision making then we need to understand the links between biodiversity and ecosystem services. We need to be aware of the limits of ecological functioning and how external pressures may impact on ecological structures and processes. Ecosystems can exhibit non-linear responses to such pressures and the possibility of rapid regime shifts”. It is then clear that the conservation of natural capital intact is the basis for the maintenance of ecosystem services at the current level. This is more and more true if we consider that we are still not completely aware about the consequences on ecosystem services providing (and consequently our well being) given by the impacts imposed to the environment. In a precautionary approach and aiming at reaching a sustainable development the natural capital should be kept at least intact.

This is way I strongly disagree with certain assumption of authors, for instance when they state:

“ Impacts to ecosystem services are potentially different than impacts to ecosystems. While impacts to ecosystems have (usually adverse) consequences for populations,species, and structure of ecosystems, impacts to ecosystem services ultimately negatively affecthuman wellbeing. For exa mple, pollution might not affect shellfish growth, but it may lead to aquaculture tenure closure for health concerns, or might affect the taste of shellfish caught at polluted sites (14). Changes to the enjoyment of shellfish aquaculture, in this case, are not a result of changes in the biophysical supply of the service but in the change to either the access to or quality of the ecosystem service (i.e. environmental impact may not mean ecosystem service” “Reframing the previous example, pollution impacts the service (the ability of people to access shellfish for food through legal restriction) or the value (the palatability of the shellfish), but the growth of shellfish (the supply) is unaffected.”

On the contrary I agree about the concept that studying or mapping ecosystem services fruition is strongly different from studying and mapping ecosystem services supply (that in the cascade coincide with ecosystem functions that represent the capability of ecosystems to provide services).

If authors want to focus on the “human side” of the cascade it is ok and in my opinion an interesting issue given that a great confusion still exist in how represent and map ecosystem services themselves. Often researches represent ecosystem services confusing them with ecosystem functions and then it is important to give tools to represent services. In my opinion the introduction should been rewritten focusing on this latter aspect rather than trying to demonstrate that ecosystem services’ supply can not be affected by modifications to natural environment.

At this purpose I suggest the reading of Burkhard B, Maes J (Eds.) (2017) Mapping Ecosystem Services. Pensoft Publishers, Sofia, 374 pp.

Methods

I suggest to list the considered ecosystem services always in the same order: these makes easier to the reader to keep them in mind

Lines 133-141 please insert a numbering list

Line 151: please briefly introduce somewhere in the text the invest model

line 154: authors should better specify what marine recreation is for instance listing all the activities composing the services as done in Supplementary materials

Line 174: should specify what to they mean with the word “stressor” and I suggest to choose only a name (activity or stressor or a third new word)

Paragraph 2.2 should be rewritten since it is unclear. Table S2 should be modified and moved to the main text: here are listed the ecosystem services and the stressor but the list of ecoservices is useless since the readier already knows it while it is not highlited the relationships between ecoservices and stressors. It would be more useful to list stressors in the first column and match each stressor with the affected ecoservices. It should be explained which indicator was used to calculate the intensity of each stressor.

Line 183-185: unclear, rephrase

Line 190-192: I should remove this sentence, as it is it seems to be useless

Line 264-267: this sentence is unclear, may be a formula would help?

Results

Results in Figure 1 and 2 are very interesting but if intensity scores are plotted on the instogram it should be explained in the figure or in the caption moreover it should be more interesting to set the upper limit of the color scale to the same value in order to allow the reader to compare the maps.

In this case a table with the higher cumulative impact reached by each ecosystem service could be added.

Lines 292-295: please add a table with these scores

Figure 3 must be better explained, it is not clear what it is plotted and its meaning. I suggest to authors to be consistent and use in the results exactly the same terminology explained in methods. If some metrics presented in results are not explained in methods authors must add an explanation.

Lines 323-327: please add table where the density of IC (see previous comment- Lines 292-295: please add a table with these scores-) and total Ic are reported

Line 327-336: not clear please rephrase: where can I see it? Not clear were this ranking is introduced. May be is this reported in appendix? In case it should be moved to results or removed at all.

Figure 4 is useless please replace it with a table.

Paragraphs 3.3 and 3.4 page 16: the numbering of paragraph is wrong: these are 3.5 and 3.6 paragraph

6. PLOS authors have the option to publish the peer review history of their article (what does this mean?). If published, this will include your full peer review and any attached files.

Reviewer #1: Yes: Ibon Galparsoro

Reviewer #2: No

---

## [Author Response · Author response to Decision Letter 0]

6 Dec 2019

We thank the editor and the two reviewer’s for their thoughtful feedback. We believe that the manuscript is now much stronger as a result of addressing the comments and helpful suggestions. Below please find our responses to the editor and reviewer comments below.

Journal Requirements:

When submitting your revision, we need you to address these additional

requirements.

Please ensure that your manuscript meets PLOS ONE's style requirements,

including those for file naming. The PLOS ONE style templates can be

found at

http://www.journals.plos.org/plosone/s/file?id=wjVg/PLOSOne_formatting_sample_main_body.pdf

and

http://www.journals.plos.org/plosone/s/file?id=ba62/PLOSOne_formatting_sample_title_authors_affiliations.pdf

Response: We have gone through the manuscript and ensured that it meets PLOS ONE’s style requirements.

1. Please include additional information regarding the survey or

questionnaire used in the study and ensure that you have provided

sufficient details that others could replicate the analyses. For

instance, if you developed a questionnaire as part of this study and it

is not under a copyright more restrictive than CC-BY, please include a

copy as Supporting Information.

Response: We have now added a paper copy of the survey as a supplement, though the survey was actually conducted online. We have also ensured that the methods of the manuscript describe the kind of information collected in the survey.

2. We note that you have stated that you will provide repository

information for your data at acceptance. Should your manuscript be

accepted for publication, we will hold it until you provide the relevant

accession numbers or DOIs necessary to access your data. If you wish to

make changes to your Data Availability statement, please describe these

changes in your cover letter and we will update your Data Availability

statement to reflect the information you provide.

Response: We still intend to submit the data to a public repository.

Reviewers' comments:

Reviewer's Responses to Questions

COMMENTS TO THE AUTHOR

1. Is the manuscript technically sound, and do the data support the

conclusions?

The manuscript must describe a technically sound piece of scientific

research with data that supports the conclusions. Experiments must have

been conducted rigorously, with appropriate controls, replication, and

sample sizes. The conclusions must be drawn appropriately based on the

data presented.

Reviewer #1: Yes

Reviewer #2: No

2. Has the statistical analysis been performed appropriately and

rigorously?

Reviewer #1: Yes

Reviewer #2: No

3. Have the authors made all data underlying the findings in their

manuscript fully available?

The PLOS Data policy [2] requires authors to make all data underlying

the findings described in their manuscript fully available without

restriction, with rare exception (please refer to the Data Availability

Statement in the manuscript PDF file). The data should be provided as

part of the manuscript or its supporting information, or deposited to a

public repository. For example, in addition to summary statistics, the

data points behind means, medians and variance measures should be

available. If there are restrictions on publicly sharing data—e.g.

participant privacy or use of data from a third party—those must be

specified.

Reviewer #1: No

Reviewer #2: No

4. Is the manuscript presented in an intelligible fashion and written in

standard English?

PLOS ONE does not copyedit accepted manuscripts, so the language in

submitted articles must be clear, correct, and unambiguous. Any

typographical or grammatical errors should be corrected at revision, so

please note any specific errors here.

Reviewer #1: Yes

Reviewer #2: No

5. Review Comments to the Author

Please use the space provided to explain your answers to the questions

above. You may also include additional comments for the author,

including concerns about dual publication, research ethics, or

publication ethics. (Please upload your review as an attachment if it

exceeds 20,000 characters)

Reviewer #1: The study presented here provides a novel method for

estimating the human activities impact on ecosystem services. This is a

very interesting and important field of research, among others because

it provides valuable knowledge that could be applied to informed

management and decision making. From the scientific point of view, the

linkages between human activities, the pressures they produce and the

associated environmental impact and their consequences in ecosystem

services provisioning, is still a very complex topic and not well-known.

Adopting the British Columbia as case study, the authors describe a

comprehensive process conducted for the obtention of the input data,

model development and implementation. In one hand, the authors have

estimated the ecosystem services production and their spatial

distribution by using real data or using already available and broadly

used models (InVEST) models for different ecosystem services). On the

other hand, authors have used information dealing with the distribution

of human activities. The most challenging point of the process is the

estimation of the impact of each activity on each of the ecosystem

services analysed. The authors have solved this by performing a

consultation to experts. Not having empirical data that link human

activities to their impacts, authors have adopted this method for

obtaining the information that afterwards was needed to develop the

models. All the methodology is exhaustively described in the

Supplementary Material.

The methodology and the results presented could be deeply discussed and

critised, but the authors are aware of the limitations and the need of

adoption of assumptions during the implementation of the approach. For

this reason, the authors dedicate a full subsection of the paper

(Section 4.5), acknowledging the weaknesses and limitations of the

approach.

Response: We thank the reviewer for their thoughtful summary of our paper, and recognition of the deep uncertainties in this kind of study and our attempt at honestly reflecting this uncertainty. We also thank the reviewer for spending the time to look through our supplementary material, in which we did try to exhaustively describe our methodology.

Specific comments

Revise and complete the affiliations.

Response: The primary author’s position has changed, and we have updated the affiliations according to these changes as well as to correspond with PLOS ONE’s requirements

I miss some more information regarding to the study site to justify why

this area was selected for the research. How is been managed at present?

Response: We have now added an extra paragraph in the methods explaining this (lines 131-150):

The coast of British Columbia, Canada spans a distance of almost 1000 km, with a complex shoreline geography of fjords, inlets, and islands extending over 25,000 km in length. It is a region of diverse resource harvesting important for ecological, economic and cultural reasons, many of which are unique to the region; for example glass sponge reefs, globally significant seabird populations, salmon, eulachon, and resident orca. The region is also important culturally for intangible benefits, including nature-based tourism. A broad range of human activities occur in this region, and a multiple cumulative impact studies have been conducted to assess impacts on the marine ecosystems (8, 10, 28). Sea-based activities include fishing, aquaculture, tourism, utility and transportation. Coastal activities also influence the marine and estuarine resources in this region, including human settlement, ports and marinas, and log storage and handling. Land-based activities occurring in the watersheds are connected to coastal marine systems through freshwater runoff and include forestry, agriculture, mining and pulp and paper mills. The region is also subject to impacts from long-range and global stressors such as climate change, pollutants and debris. Activities that include vessel use additionally include the stressors associated with either small or large vessel use in their cumulative risk. Management of coastal British Columbia is siloed, with sea-based activities under the purview of Fisheries and Oceans Canada, land-based resources under provincial authority (Forest, Lands, Natural Resource Operations and Rural Development), coastal national parks under Parks Canada, and Environment and Climate Change Canada, and towns and human settlements often governed by local governments. Because of the diverse natural resources and ecosystem services, as well as the past research done on ecological impacts, we chose to study this region as a case study to study cumulative impacts on ecosystem services.

Additional information such as the coastal length, surface or area of

the case study, depth ranges. A very short description. A couple of

lines would be enough.

Response: We have added some information on that in the inserted paragraph. We have added “The coast of British Columbia, Canada spans a distance of almost 1000 km, with a complex shoreline geography of fjords, inlets, and islands extending over 25,000 km in length.” (lines 131-132).

Line 166. “Bathymetry and topography was used to calculate

viewshed”. I would say that what is below the sea surface can not be

seen and that is not part of the viewshed.

Response: We agree with the reviewer. We have removed reference to bathymetry, but have added that our model does include the curvature of the earth.

Lines 272-274, would need a rephrasing to make it more clear.

Response: We agree with the reviewer that this sentence wasn’t clear. We have updated it to read (lines 310-315):

Cumulative impacts were calculated twice: first, cumulative impact scores were calculated without the service and value dimensions; next, cumulative impact scores were calculated with service and value dimensions. The difference between these two calculations reveals the contribution of considering the service and value dimensions when assessing cumulative impacts on ecosystem services.

Section 4.5. As stated previously, this is an important section of the

paper because the authors list all the limitations and assumptions that

they had to adopt during the modelling process. Nevertheless, I still

miss one more point. The authors are assuming that the ecosystem

services production and the location of the maritime activity overlap.

This is an assumption needed when operating between different

information layers, but I think that this should be also discussed.

Certain activities could have an effect on ecosystem services that are

produced elsewhere. For example, fishing activity (e.g. bottom

trawling), could impact nursery grounds of species of commercial

interest that are fished in other locations. Thus, the impact on the

ecosystem service and the activity are not spatially coincident. The

same happens for other ecosystem services. This is a complex point of

ecosystem functioning but that it would be interesting to mention in

this section.

Response: The reviewer raises an interesting point about the static nature of the model and its basis in physical overlap. The current methods do address this concern somewhat, but not fully: e.g., our model incorporates an area of influence from activities based on prominent stressors from each activity. We stated so in the methods (lines 226-227), “This dataset considers the area of influence of each human activity, with the extent of each area of influence dependent on prominent stressors associated with each activity.” We also noted that the risk scoring includes a component on “community extent” which attempts at addressing some of the ecological dynamics that the reviewer raises.

However, we agree with the reviewer that this is still a limitation, particularly because we have represented ecosystem services (especially biologically based ecosystem services) largely from sites of human harvest. We have added the following to the discussion (lines 622-628): 

As a simple model of cumulative impacts we also could not explicitly represent some important ecological dynamics and cascading effects that do not co-occur spatially. For example, bottom trawling can negatively affect nursery grounds for species that are fished in other locations. While we tried to capture some of these dynamics by asking experts to score the “community extent” of risk, we cannot capture the full complexity of human impact on ecosystem function, particularly where impacts affect habitats and ecosystem functions underpinning services that occur elsewhere.

Figure 1. I think that it would be interesting to add labels to the axis

of the bar charts.

Response: We have now added axis labels and legends to this figure.

Figure 2. It would be interesting to add a graphical legend indicating

what are representing the red and grey graphs.

Response: We have now added axis labels and legends to this figure.

Figure 3. Add axis labels.

Response: We have added axis labels and a legend to this figure.

I have added some extra comments in the Supplementary Material.

Response: We thank the reviewer for their thorough review. We have made modifications to the reviewer’s queries. First, we have accepted the reviewer’s changes to details in the tables. Second, regarding the scenarios used in the probabilistic inversion method, the method selects “scenarios” based on the probability distribution of ranks for exposure and consequence criteria (based on expert input), which have a vast number of possible combinations. We have added the following to the supplemental methods: “Probabilistic inversion generates scenarios by selecting from the probability distribution of exposure and consequence criteria ranks (based on expert the distribution of expert responses), which allows for thousands of nonexclusive combinations.”

Finally, for the reviewer’s question about the coastal vulnerability to coastal erosion, the vulnerability classes were not determined by us but are standard vulnerability data collected from GeoBC. While the reviewer asks whether a narrow sand and gravel beach should be assigned a higher vulnerability of erosion, we note that both the wider and narrow sand beaches on rock ramps have the highest erosion vulnerability scores. It is the presence of sand on rock ramps that makes for high vulnerability. The reviewer may still be right that the narrower beach should have higher vulnerability, however they both are at the highest index of vulnerability and the index does not allow for comparison within the highest index class

Reviewer #2: The aim of the paper is very good and the subject is novel,

very interesting and very topical. Nonetheless results are presented in

a confused way: the main problem in my opinion resided in the fact taht

authors wanted to convey to much information. Some results have then

been reported in supplementary materials while several results

paragraphs deals with the same figures. As a consequence the reader is

not allowed to a full understanding.

Response: We thank the reviewer for their encouragement on our aims. While we acknowledge that there is a lot of information presented, we do not think it is “too much”. We thank the reviewer for their recommendation on presenting information more clearly, and we have aspired to improve this in several ways described below and above.

I suggest to reorganise the paper and

1. to be consistent and to use the same terms in methods and results

Response: We have gone through the manuscript and ensured this consistency. First, we have clarified important, but related terms. We differentiate between “human activities” and “environmental stressors”, the latter as “processes that impact the environment from either human activities or long term change” (line 207). We also distinguish between risk and impact, with the former defined as “the potential of an activity to impact an ecosystem service where they co-occur” (lines 154-155).

We have also ensured that we are consistent in using these terms throughout the methods and results. The marked-up version of the paper will show some instances of changing terms of “drivers and activities” to “activities and stressors”, and “future impact” to “future risk”.

2. it would be useful to have the same sections in methods and results

and in the same order of appearance

Response: We have gone through and ensured alignment between methods and results. Importantly, where possible we have used the same header language between the methods and results now. Note however since our methods explain the risk evaluation and mapping across multiple steps, we cannot have the same sections in the methods and results, because multiple sections of the methods will have produced a single section of results.

3. choose most important results and information and focus on them

better explaining how they are obtained

Response: We thank the reviewer for their comments, but without a clear idea of what they do not understand or what they would like explained better, we do not know if we have addressed the reviewer’s concern. Regardless we have gone through the paper and ensured that we describe results. Importantly, we believe we have highlighted the most important results.

I made comments until the results section, for discussion i would like

to see a revised verison of the paper

Response: We thank the reviewer for their comments, but without specific comments on the discussion we cannot be certain we are addressing their concerns. However, we have revised the paper from the introduction to the discussion and conclusion to ensure consistency and in reference to the comments of Reviewers 1 and 2.

Here below my specific comments: please pay particular attention to

comments related to the intro section

Introduction

I appreciate that authors cite the ecosystem services cascade that

represents, in my opinion, a turning point concerning the ecosystem

services theory. Nonetheless I am not sure to share their interpretation

about the cascade.

The cascade framework was developed in order to highlight the dependence

of human well-being to ecosystems. As the authors (Haines-Young, R., &

Potschin, M. (2010). The links between biodiversity, ecosystem services

and human well-being. Ecosystem Ecology: a new synthesis, 1, 110-139.)

of the cascade say “Ecologists will increasingly have to work

alongside economists, geographers and a range of other social scientists

to understand the value that biodiversity and ecosystem services have,

to assess the costs and benefi ts of different conservation and

management strategies, and to help design the new governance systems

needed for sustainable development. Biodiversity has intrinsic value and

should be conserved in its own right. However, the utilitarian arguments

which can be made around the concept of ecosystem services and human

well-being are likely to become an increasingly central focus of future

debates about the need to preserve ‘natural capital’. The wider

research community needs to engage in such debates. Although long-term

sustainable development has come to mean many things, the concept must

include the maintenance of ecosystem services and the elements of human

well-being that depend upon healthy ecosystems. If the Ecosystem

Approach is to be embedded in decision making then we need to understand

the links between biodiversity and ecosystem services. We need to be

aware of the limits of ecological functioning and how external pressures

may impact on ecological structures and processes. Ecosystems can

exhibit non-linear responses to such pressures and the possibility of

rapid regime shifts”. It is then clear that the conservation of

natural capital intact is the basis for the maintenance of ecosystem

services at the current level. This is more and more true if we consider

that we are still not completely aware about the consequences on

ecosystem services providing (and consequently our well being) given by

the impacts imposed to the environment. In a precautionary approach and

aiming at reaching a sustainable development the natural capital should

be kept at least intact.

This is way I strongly disagree with certain assumption of authors, for

instance when they state:

“ Impacts to ecosystem services are potentially different than impacts

to ecosystems. While impacts to ecosystems have (usually adverse)

consequences for populations,species, and structure of ecosystems,

impacts to ecosystem services ultimately negatively affecthuman

wellbeing. For exa mple, pollution might not affect shellfish growth,

but it may lead to aquaculture tenure closure for health concerns, or

might affect the taste of shellfish caught at polluted sites (14).

Changes to the enjoyment of shellfish aquaculture, in this case, are not

a result of changes in the biophysical supply of the service but in the

change to either the access to or quality of the ecosystem service (i.e.

environmental impact may not mean ecosystem service” “Reframing the

previous example, pollution impacts the service (the ability of people

to access shellfish for food through legal restriction) or the value

(the palatability of the shellfish), but the growth of shellfish (the

supply) is unaffected.”

On the contrary I agree about the concept that studying or mapping

ecosystem services fruition is strongly different from studying and

mapping ecosystem services supply (that in the cascade coincide with

ecosystem functions that represent the capability of ecosystems to

provide services).

If authors want to focus on the “human side” of the cascade it is ok

and in my opinion an interesting issue given that a great confusion

still exist in how represent and map ecosystem services themselves.

Often researches represent ecosystem services confusing them with

ecosystem functions and then it is important to give tools to represent

services. In my opinion the introduction should been rewritten focusing

on this latter aspect rather than trying to demonstrate that ecosystem

services’ supply can not be affected by modifications to natural

environment.

At this purpose I suggest the reading of Burkhard B, Maes J (Eds.)

(2017) Mapping Ecosystem Services. Pensoft Publishers, Sofia, 374 pp.

Response: We thank the reviewer for this thoughtful feedback. We completely agree with the reviewer that ecosystem services are modified by the natural environment. We realize that stating that “impacts to ecosystem services are potentially different than impacts

to ecosystems” can be read as saying that impacts on ecosystems are not impacts on ecosystem services, and so we have removed that statement and clarified our point. We did not mean to suggest that environmental change does not affect ecosystem services, but rather that in order to understand impacts to ecosystem services, we need to think beyond only thinking about environmental changes. We emphasize this when we write: “While any human activity that impacts ecosystems has the potential to impact ecosystem services, ecosystem services includes dynamics beyond the ecological production of potential benefits to people” (lines 61-63). However, we can see how the previous text was unclear on this point and have modified text in the introduction to clarify our intended meaning. On a further point, while we do recognize the work of Haines-Young, R., & Potschin, M. (2010) on the concept of ecosystem service cascades, we rather operationalize a more recent version of the cascade framework, by Tallis et al. (2012). This more recent framing is explicitly focused on the interplay between social and biophysical dimensions of ecosystem services, as is the focus of our work. Finally, note that—contrary to the reviewer’s comment—our work extends well beyond the “human side” of the cascade, since much of our modeling work involves both the biophysical basis of ecosystem services. We have changed text in the introduction to read as follows (lines 63-73): 

Assessing impacts to ecosystem services therefore must include considerations beyond changes to the natural environment. The dynamic interactions between environmental change and human beneficiaries of ecosystem services may in some cases mean that ecosystem services are degraded without substantial impacts on the underlying environment, and in other cases that the underlying environment is degraded without substantial impact on ecosystem services. Differences in how ecosystems and ecosystem services are impacted is largely unexplored in the literature, however. For example, pollution might minimally affect shellfish growth, but it might lead to aquaculture tenure closure for health concerns, or affect the taste of shellfish caught at polluted sites (14). Changes to ecosystems’ contribution to shellfish aquaculture, in this case, are not a result of changes in the biophysical supply of the service. Rather, the ecosystem service is impacted via either the access to or quality of the service. 

Methods

I suggest to list the considered ecosystem services always in the same

order: these makes easier to the reader to keep them in mind

Response: We have gone through the manuscript and made sure that lists of ecosystem services are presented in the same order. 

Lines 133-141 please insert a numbering list

Response: We have now made this a numbered list

Line 151: please briefly introduce somewhere in the text the invest

model

Response: We have now added the following text (lines 180-183): “InVEST (integrated valuation of ecosystem services and tradeoffs) is a decision-support tool for mapping and valuing ecosystem services, that generates spatially explicit models of ecosystem services based on underlying ecosystem characteristics.”

line 154: authors should better specify what marine recreation is for

instance listing all the activities composing the services as done in

Supplementary materials

Response: We have now added the following text (line 186-187): “Marine recreation includes kayak, recreational boating, recreational fishing, and populous sites for recreation, including camping and dive sites.”

Line 174: should specify what to they mean with the word “stressor”

and I suggest to choose only a name (activity or stressor or a third new

word)

Response: We specifically differentiate activities and stressors because we include climate change risk, and erosion, which are processes not associated with a single human activity. We have chosen to define the term as “processes that impact the environment from either human activities or long term change” (line 207).

Paragraph 2.2 should be rewritten since it is unclear.

Response: We have tried clarifying the section but we cannot be certain it satisfies the reviewer (changes can be viewed in the marked-up version of the manuscript). We would be happy to make additional revisions 

based on further guidance.

 Table S2 should

be modified and moved to the main text: here are listed the ecosystem

services and the stressor but the list of ecoservices is useless since

the readier already knows it while it is not highlited the relationships

between ecoservices and stressors. It would be more useful to list

stressors in the first column and match each stressor with the affected

ecoservices. It should be explained which indicator was used to

calculate the intensity of each stressor.

Response: We have moved the table to the main text. We agree with the reviewer on the change to the structure and have done so here. However, given the complexity of the table with including the indicator (all ecosystem services have multiple activities and stressors associated), it would be too difficult to attach indicators. We have left this information for the supplement

Line 183-185: unclear, rephrase

Response: We have rephrased the sentence to read, “We treat ecosystem services as broad categories (such as demersal vs pelagic fisheries) and activities and stressors that cause impact as specific categories because experts indicated that broad types of ecosystem services (such as various benthic fisheries, or various pelagic fisheries) are impacted in similar ways, while they indicated that they did not treat human activities and stressors in a similar way. ” (lines 215-219).

Line 190-192: I should remove this sentence, as it is it seems to be

useless

Response: We disagree with the reviewer here but would be interested to better understand the reviewer’s thinking. In our opinion, the finding by Ban et al (2010) is useful methodologically, and explains why we chose to treat our data the way we did. As we write, “the number of data layers influences the overall cumulative impact scores (8), and we did not want to overly bias impact based on fisheries scores” (lines 224-226).

Line 264-267: this sentence is unclear, may be a formula would help?

Response: We have clarified this sentence to read, “All intensity data for human activities were log transformed and normalized by dividing by the largest intensity value found for each activity and stressor across the BC coast to generate a dimensionless 0-1 intensity scale” (lines 302-304).

Results

Results in Figure 1 and 2 are very interesting but if intensity scores

are plotted on the instogram it should be explained in the figure or in

the caption moreover it should be more interesting to set the upper

limit of the color scale to the same value in order to allow the reader

to compare the maps.

Response: We have added more description to the plots (something that the first reviewer also mentioned) and have ensured that the caption also describes that the figure shows the maps and the total impact. Note that some ecosystem services register total impact values an order of magnitude higher than other ecosystem services, and so putting all graphs on the same upper limit would make some graphs very hard to read.

In this case a table with the higher cumulative impact reached by each

ecosystem service could be added.

Response: We are not sure what the reviewer means by this statement. Does the reviewer suggest Add adding a table ranking the activities and stressors for each ecosystem service by how much impact they contribute? We feel that would be largely redundant with figures 1 and 2, and the text in the results, which already showcase the cumulative impact scores by activity and stressor, and provides cumulative impact scores.

Lines 292-295: please add a table with these scores

Response: We have considered it and decided that adding another table would be redundant with figure 3 (a redundancy that the journal has stated we should avoid), so we have retained the ranked list.

Figure 3 must be better explained, it is not clear what it is plotted

and its meaning. I suggest to authors to be consistent and use in the

results exactly the same terminology explained in methods. If some

metrics presented in results are not explained in methods authors must

add an explanation.

Response: We have recreated the figure with more explanation of the axes, something reviewer 1 also mentioned.

Lines 323-327: please add table where the density of IC (see previous

comment- Lines 292-295: please add a table with these scores-) and total

Ic are reported

Response: As noted previously, we believe these tables would be largely redundant with information already presented in figures. We have decided not to add this table.

Line 327-336: not clear please rephrase: where can I see it? Not clear

were this ranking is introduced. May be is this reported in appendix? In

case it should be moved to results or removed at all.

Response: The ranking is discussed in the text. We have added the word “ranked” at the beginning of this section so it is further clarified. We have clearly stated where we rank for both total impact scores and per-cell impact.

Figure 4 is useless please replace it with a table.

Response: While we respect the reviewer`s preference for a table, we disagree that figure 4 is “useless” as it clearly showcases trends. We have workshopped how to represent this data and found this type of graph best for showcasing the results.

Paragraphs 3.3 and 3.4 page 16: the numbering of paragraph is wrong:

these are 3.5 and 3.6 paragraph

Response: We thank the reviewer for pointing this out. We have fixed the numbering

---

## [Decision Letter · Decision Letter 1]

14 Jan 2020

PONE-D-19-18924R1

Mapping cumulative impacts to coastal ecosystem services in British Columbia

PLOS ONE

Dear Dr. Singh,

Thank you for submitting your manuscript to PLOS ONE. After careful consideration, we feel that it has merit but does not fully meet PLOS ONE’s publication criteria as it currently stands. Therefore, we invite you to submit a revised version of the manuscript that addresses the points raised during the review process.

The reviewer lamented that the authors did not take in due account their previous comments. I want to offer the authors the opportunity to reconsider once and forever their ms in the light of the reviewer’s renewed comments. I will ask gain the reviewer to check the modifications introduced by the authors. Should the reviewer be again dissatisfied, the paper would be rejected. 

*We would appreciate receiving your revised manuscript by Feb 28 2020 11:59PM. When you are ready to submit your revision, log on to https://www.editorialmanager.com/pone/ and select the 'Submissions Needing Revision' folder to locate your manuscript file*.

*If you would like to make changes to your financial disclosure, please include your updated statement in your cover letter*.

To enhance the reproducibility of your results, we recommend that if applicable you deposit your laboratory protocols in protocols.io, where a protocol can be assigned its own identifier (DOI) such that it can be cited independently in the future. For instructions see: http://journals.plos.org/plosone/s/submission-guidelines#loc-laboratory-protocols

We look forward to receiving your revised manuscript.

Kind regards,

Carlo Nike Bianchi

Academic Editor

PLOS ONE

Reviewers' comments:

Reviewer's Responses to Questions

**Comments to the Author**

1. If the authors have adequately addressed your comments raised in a previous round of review and you feel that this manuscript is now acceptable for publication, you may indicate that here to bypass the “Comments to the Author” section, enter your conflict of interest statement in the “Confidential to Editor” section, and submit your "Accept" recommendation.

Reviewer #2: (No Response)

2. Is the manuscript technically sound, and do the data support the conclusions?

Reviewer #2: No

3. Has the statistical analysis been performed appropriately and rigorously? 

Reviewer #2: I Don't Know

4. Have the authors made all data underlying the findings in their manuscript fully available?

Reviewer #2: (No Response)

5. Is the manuscript presented in an intelligible fashion and written in standard English?

Reviewer #2: No

6. Review Comments to the Author

Reviewer #2: I did not find the modification made by authors completely effective: most part of my comment have been ignored, hopefully since authors did not completely understand them. The paper, to me, cannot be published as it is and still need a strong revision and several modifications to be published. Here below my new comments, I tried to be clearer. I suggest the authors to put a diagram of their theoretical path in methods to clarify: aims, methods, results and re-arrange the paper accordingly.

Introduction

I made a long comment to explain the authors that some sentences about the interactions among services and environment should have been changed buit they ignored it. As a consequence I strongl suggest to remove lines from 68 to 78 “Assessing….service”(version of the paper with track changes). Analogously lines from 90 to 98 (“Reframing...an impact to an ecosystem service”) must be removed: they are disorganized and misleading. I suggest the authors to find in literature some theoretical example of the information they want to convey. Please put references when reporting this. Modifications made do not solve the issue.

I also suggested to cite Burkhard B, Maes J (Eds.)

(2017) Mapping Ecosystem Services. Pensoft Publishers, Sofia, 374 pp. But it has not been done

They say that their approach is based on Tallis et al. 2012 (moreover they theory they are referring to is better explained, in my opinion in Tallis et al. 2012, “A Global System for Monitoring Ecosystem Service Change”, rather than in “New metrics for managing and sustaining the ocean's bounty”) but they discuss Heines-Young. As a consequence I strongly suggest to better refer to Tallis et al. (2012) and to deepen their theory without referring to the cascade.

The Introduction must then be rewritten, as I already suggested.

Lines 122-128: unclear, please rephrase

Lines 132-133: the authors say that their aim is to know Which ecosystem services face the greatest cumulative impact in coastal British Columbia but the reader still don’t know what they mean with cumulative, what does it means that “the ecosystem services face the greatest”

The underlying theory must be explained more clearly

Line 157: not clear to me what the authors mean with “siloed”

Methods

I suggested to reorganise and be consistent. Authors list 5 phases: since subsequently 5 subparagraphs are present I strongly suggest to name each paragraph accordingly to each phase listed

e.g. if the paragraph 2.3 is “spatial representation of ecosystem services” in the list at line 172 it should be written: “spatial representation of ecosystem services: we mapped eight ecosystem services…..”

Line 189: it is not clear which variables have been mapped, please inserted a table with very brief description for each service or insert an explanation in the text

Lines 191: please list the services here so move here lines from 195 to 198.

If I correctly understand only renewable energy and aquaculture were modelled without using INVEST. It can be easily said, at lines 191-192: the eight services were all modelled using InVest excluding energy and aquaculture for which we used the publicly available spatial.

Still not clear to me the difference between services and human activities or stressor (please choose a unique term).

I think these definition should be given with a very clear example of each definition

So please define and give an example of:

- ecosystem services

- impacting activities

- Risk

I know they state they defined these in the text but definitions are sparse and not clear. Moreover it is confusing to me that fisheries are both a service and a stressor, this should be clarified.

Lines 250-253: this should be in some way explained before when the concept of cumulative impact is introduced.

Results

In my opinion results should be simplified and better presented

First, authors should avoid to list indicator values in text as list, please use table. Moreover presenting impact scores were calculated before without the service and value dimensions and later with service and value dimensions makes difficult to read the paper. I suggest to remove this or to present it in clearer way, for example with two separate sessions. Moreover as I already told authors should better explain in method what are these “dimensions”: supply dimensions, values dimensions...there are too many concepts and terms and this confusing. As I already suggest in the first review is necessary to focus on some main results they want to convey and present them in a clearer and synthetic way.

Moreover I think that the part regarding InVEST, mapping and the cumulative index should be clearly separated in both methods and results and merged when put together to make the cumulative impact. I think this should be a good framework for the entire paper:

-maps

- expert

-cumulative measures

Lines 350-355 should be put in a table

Line 355: is the supply dimension identified by identified by spatial representation of ecosystem services? Please be consistent: it must be clear to which part of the methods the authors refer in results

Lines 357-381: need table with Ic

Line 382: how is it calculated the “total summed impact”, insert also these values in a table. It is not necessary to list all values in the text and this makes the reading very hard. If values are diffent for only supply dimension and not please insert both

Lines 391:where can I see this? Fig. 1? If yes please cite here

Lines 421-430: again, these results should be presented with a table, directly compared with the previous ones. Please name the different Ic with different names: the reader reads all along the text Ic but if I correctly understand this sometimes refers to the index with value dimension, sometimes not.

Figure 6: it should be better explained in methods how these values are obtained

7. PLOS authors have the option to publish the peer review history of their article (what does this mean?). If published, this will include your full peer review and any attached files.

Reviewer #2: No

---

## [Author Response · Author response to Decision Letter 1]

27 Feb 2020

We thank the editor and the reviewer for their feedback and willingness to clarify points raised. Without this further clarification we believe we would not have been able to properly address the reviewer concerns. We have taken this opportunity to clarify our manuscript throughout and ensure we are consistent with our terms and definitions. Below you will find our responses to the particular reviewer comments.

Reviewer #2: I did not find the modification made by authors completely effective: most part of my comment have been ignored, hopefully since authors did not completely understand them. The paper, to me, cannot be published as it is and still need a strong revision and several modifications to be published. Here below my new comments, I tried to be clearer. I suggest the authors to put a diagram of their theoretical path in methods to clarify: aims, methods, results and re-arrange the paper accordingly.

Response: As we have been in contact with the reviewer following this review, it became clear that there were misunderstandings on both sides. We thought that we had responded to all of the reviewer’s concerns in the last round, but the reviewer thought that we ignored them. It has become apparent to us that we did not understand the reviewer’s comments in full, and so our responses did not fully address the reviewer’s comments. We have now further clarified our framing of ES theory and removed sections that might cause misunderstandings. There may still be some disagreement over ES theory, so we provide extra examples and literature to back up our points in this response.

We agree with the reviewer that having a diagram to give a high-level approach to our study is useful. We have now included such a diagram (Fig 1).

Reviewer: 

Introduction

I made a long comment to explain the authors that some sentences about the interactions among services and environment should have been changed buit they ignored it. As a consequence I strongl suggest to remove lines from 68 to 78 “Assessing….service”(version of the paper with track changes). Analogously lines from 90 to 98 (“Reframing...an impact to an ecosystem service”) must be removed: they are disorganized and misleading. I suggest the authors to find in literature some theoretical example of the information they want to convey. Please put references when reporting this. Modifications made do not solve the issue.

Response: Based on further exchanges with the reviewer after receiving this feedback, we think we now understand the reviewer’s perspective and have addressed it. Our primary point here was that impacts to ecosystem services can be social in nature (via access and demand/perceived quality), which may add impact on top of biophysical changes. Theoretically, there can also be cases where changes to people’s access and demand/perceived quality of ecosystem services impacts ecosystem services without undermining the biophysical provision. We have clarified with the reviewer via email that his/her primary concern was that it is important to be precautionary and not suggest that changes to ecosystem services do not imply changes to the underlying ecosystem. We do not wish to convey that impacts to the underlying ecosystem do not impact ecosystems services (we think they usually do, and they always affect the potential of future ecosystem service enjoyment), and so we have simplified and clarified the text here. 

For further clarification of our perspective (only if this is needed), here’s more detail with examples. While ecosystem service supply is always important because you cannot have ecosystem services without their supply, this does not mean that all impacts to ecosystem service as benefits are mediated by supply. In perhaps the most famous example of ecosystem service change being driven by changes in access and preferences (and not the ecosystem), pollinating services for coffee in Costa Rica first decreased as coffee prices dropped then were rendered worthless as farmers switched from coffee to pineapple (a more valuable crop which is wind pollinated and not animal pollinated). The potential for pollination services was still there (if in the future animal pollinated crops were planted again), but because of economic choices the value of pollinators decreased (McCauley 2006). Other examples of ecosystem services changed by human access and value exist, and some are provided right in the literature we refer to (e.g. Tallis et al. 2012). Examples from Tallis et al. 2012 include that shoreline protection from kelp and the provision of wave energy as ecosystem services are regulated by the existence and location of human built infrastructure. Changes to the location and existence of this infrastructure will change the ecosystem services being delivered to people without necessarily affecting the underlying ecosystem that is providing the potential service.

The reviewer has also strongly suggested that we refer to a theoretical example, and so to avoid more disagreements about the example we previously provided about shellfish production, we have instead relied on examples that we document in other peer reviewed publications (Singh et al. 2017). Our text now reads (lines 64-75):

Any human activity that impacts ecosystems has the potential to impact ecosystem services in multiple ways. In addition to impacts on the biophysical production of services, human activities and infrastructure can also undermine the “consumption” of ecosystem services [14].That is, a human activity can undermine people’s ability to access or enjoy an ecosystem service. The role of impacts to the production versus the consumption of ecosystem services is largely unexplored in the literature, however. For example, in New Zealand shellfish aquaculture sites and shipping lanes can limit commercial fishing operations in an area because of legislation that limits their overlap, impacting the contribution of fisheries ecosystem services [15]. In this case, the assessed impact of shipping and aquaculture on fisheries operated through changes in access and not through impacts on biophysical supply (though the effluent from increased shipping may impact biophysical supply in the long term).

We have also changed the text revisiting the example in the framework of ecosystem service supply, service, and value, to read (lines 83-93):

Impacts to ecosystem services can be characterized at each step in the ecosystem services ‘cascade’ [13], with impact drivers potentially affecting supply (the biophysical components that produce ecosystem services), service (the ability of people to access and benefit from a service), and value (people’s preferences for ecosystem services, 13). Reframing the previous example, shipping lanes and aquaculture sites impact the service (the ability of people to access fisheries for food through legal restriction), even if the growth and availability of fish (the supply) might be unaffected. In this case (where shipping lanes and aquaculture restrict fisheries) what might not be considered an environmental impact (to fish) would be considered an ecosystem service impact. While this cascade is useful for parsing out the dynamics of impacts to ecosystem services, the relative importance of these factors (supply, service, and value) in regulating impact to ecosystem services is not known.

McCauley DJ (2006) Selling out on nature. Nature 443: 27-28.

Singh GG, Sinner J, Ellis J, Kandlikar M, Halpern BS, et al. (2017) Mechanisms and risk of cumulative impacts to coastal ecosystem services: An expert elicitation approach. Journal of environmental management 199: 229-241.

Tallis H, Lester SE, Ruckelshaus M, Plummer M, McLeod K, et al. (2012) New metrics for managing and sustaining the ocean's bounty. Marine Policy 36: 303-306.

Reviewer: I also suggested to cite Burkhard B, Maes J (Eds.)

(2017) Mapping Ecosystem Services. Pensoft Publishers, Sofia, 374 pp. But it has not been done

Response: We have now added this citation

Reviewer: They say that their approach is based on Tallis et al. 2012 (moreover they theory they are referring to is better explained, in my opinion in Tallis et al. 2012, “A Global System for Monitoring Ecosystem Service Change”, rather than in “New metrics for managing and sustaining the ocean's bounty”) but they discuss Heines-Young. As a consequence I strongly suggest to better refer to Tallis et al. (2012) and to deepen their theory without referring to the cascade.

Response: As we have noted in our previous contact with the reviewer, the reviewer is suggesting that we cite a different paper, one that actually cites the paper that we cite as the primary and most comprehensive source of the theory. Accordingly, we have retained our references to Tallis et al (2012) “new metrics for sustaining the ocean’s bounty”. We have also added the citation to the paper the reviewer suggests, however, because it is a paper based on monitoring ES change, which is also relevant to our paper. Our responses above address our attempts to further address the theory put forward by the literature. Finally, given the further exchanges with the reviewer, we believe that both sides (us and the reviewer) see the ecosystem cascade theory as well as that proposed by Tallis et al (2012) to be largely complementary.

Reviewer: The Introduction must then be rewritten, as I already suggested.

Response: The introduction has been revisited and rewritten again now to address the points brought up by the reviewer.

Reviewer: Lines 122-128: unclear, please rephrase

Response: We agree that these lines were not as clear as they could have been. I have updated the following lines for clarity, focusing on how linking the ideas of ecosystem service supply, service, and value dimensions requires a better causal understanding of impacts. The text now reads (lines 114-117): 

However, changes to people’s access, use, and perceived quality of service may also be important for understanding impacts to ecosystem services [27,28], and understanding the mechanism of impacts on ecosystem services can help address management goals [17].

Reviewer: Lines 132-133: the authors say that their aim is to know Which ecosystem services face the greatest cumulative impact in coastal British Columbia but the reader still don’t know what they mean with cumulative, what does it means that “the ecosystem services face the greatest”

The underlying theory must be explained more clearly

Response: We agree that we can better describe what “cumulative” impacts are. We have now added an explicit definition for cumulative impacts in the opening paragraph (lines 50-53):

The need to understand and manage simultaneous impacts of multiple human activities on ecosystems (such as fisheries and agricultural runoff impacting fish habitat concurrently), referred to here as cumulative impacts, has led to widespread uptake in cumulative impact mapping methods around the world [3,4,5,6,7,8,9,10,11].

We also explicitly state how we define cumulative impacts with regards to our study (line 124127): “This work, alternately, does so for ecosystem services themselves, representing cumulative impact as the combined total impact that an ecosystem service experiences from a variety of co-occurring drivers of impact (such as ocean acidification, agricultural runoff, and fishing)”. This framing largely adapts the theory from Halpern et al (2008, 2015). 

We have also changed the language away from “the greatest” impact to “the most severe impact”, which we think clarifies the concept, especially in the context of our definition of the combined total impact that ecosystem services experience. 

Halpern BS, Walbridge S, Selkoe KA, Kappel CV, Micheli F, et al. (2008) A Global Map of Human Impact on Marine Ecosystems. Science 319: 948-952.

Halpern BS, Frazier M, Potapenko J, Casey KS, Koenig K, et al. (2015) Spatial and temporal changes in cumulative human impacts on the world’s ocean. Nature communications 6: 1-7.

Reviewer: Line 157: not clear to me what the authors mean with “siloed”

Response: By “siloed” we refer to the notion that managing different ecosystems and industries is done in a piecemeal way, so that, for example fisheries and agriculture are managed independently with little to no coordination between regulating bodies. We have changed this text to replace “siloed” with “in a piecemeal way” 

Reviewer: 

Methods

I suggested to reorganise and be consistent. Authors list 5 phases: since subsequently 5 subparagraphs are present I strongly suggest to name each paragraph accordingly to each phase listed

e.g. if the paragraph 2.3 is “spatial representation of ecosystem services” in the list at line 172 it should be written: “spatial representation of ecosystem services: we mapped eight ecosystem services…..”

Response: We have now added the header to begin each line that introduces each phase, as suggested by the reviewer

Reviewer: Line 189: it is not clear which variables have been mapped, please inserted a table with very brief description for each service or insert an explanation in the text

Response: We have a supplemental table with all the variables used. We have reorganized this table so that the variables are associated with each relevant ecosystem service (S2 Table). We have referred to this table in the text and we have added brief description of each ecosystem service in the text (lines 204-214):

Coastal aesthetics was modeled by calculating the viewshed from kayaking, recreational boating, population centers, recreational fishing. Coastal protection was modeled assessing the protection provided by marine vegetation (kelp and seagrass) to different types of shoreline (sandy to rocky). Benefits from commercial demersal fisheries and benefits from commercial pelagic fisheries were modeled by aggregating multiple commercial fishery spatial data layers. Coastal recreation includes kayak, recreational boating, recreational fishing, and populous sites for recreation, including camping and dive sites. We modeled “potential” energy generation because British Columbia currently does not have wave and tidal energy operations, but there is interest in harnessing this energy supply. Benefits from finfish and shellfish aquaculture were modeled by aggregating spatial data of finfish and shellfish aquaculture. For more detail on the ecosystem service models, see Supporting Methods.

Reviewer: Lines 191: please list the services here so move here lines from 195 to 198.

Response: We have moved these lines up to line 191 (now lines 197-201)

Reviewer: If I correctly understand only renewable energy and aquaculture were modelled without using INVEST. It can be easily said, at lines 191-192: the eight services were all modelled using InVest excluding energy and aquaculture for which we used the publicly available spatial.

Response: We have now stated the above, as the reviewer mentions

Reviewer: Still not clear to me the difference between services and human activities or stressor (please choose a unique term).

Response: Services refer to ecosystem services, while human activities or stressors refer to the causes of impact. We use these terms because they are widely used in the cumulative impacts literature (Murray et al 2016, Singh et al, 2017). We have changed the term to “drivers” to capture the activities and stressors (also a term used in cumulative impacts literature) and have made sure we use the term consistently throughout.

We have expanded on our definitions of ecosystem services, drivers, and impacts. We have added the following definitions: 

“Ecosystem services are the environmental processes that render benefits to people. Implicit to this definition is that, while ecosystem functions are essential for providing ecosystem services, these services do not exist without human beneficiaries [13,14].” (lines 62-64).

“We define drivers as the human activities and long-term stressors (such as ocean acidification) that contribute to a deterioration of benefits derived from ecosystem services. We define stressors as the processes that undermine ecosystem service benefits, and we define impacts as the deterioration of ecosystem service benefit. For example, agriculture contributes to runoff that can lead to sedimentation which can smother shellfish harvested by people [15,16]. In this example, agriculture is a driver, sedimentation is a stressor, and reduced shellfish biomass for food is the impact [15,17].” (lines 77-83)

Murray CC, Mach ME, Martone RG, Singh GG, O M, et al. (2016) Supporting risk assessment: accounting for indirect risk to ecosystem components. PLoS ONE 11: e0162932.

Singh GG, Sinner J, Ellis J, Kandlikar M, Halpern BS, et al. (2017) Mechanisms and risk of cumulative impacts to coastal ecosystem services: An expert elicitation approach. Journal of environmental management 199: 229-241.

Reviewer: I think these definition should be given with a very clear example of each definition

So please define and give an example of:

- ecosystem services

- impacting activities

- Risk

I know they state they defined these in the text but definitions are sparse and not clear. 

Response: We have clarified these definitions in the text. See the above definition for definitions of ecosystem services, drivers (formerly impacting activities) and impacts. We have always had a definition of risk in the manuscript, which we state is “the potential of a driver to impact an ecosystem service where they co-occur”. We have also further clarified our definition of risk, adding the following to the section describing the cumulative impacts model: “We define risk as the potential of a driver to impact a particular ecosystem service. In the context of our cumulative impact model, risk is the potential of a single event of an activity to impact a given ecosystem service.” (lines 266-268).

Reviewer: Moreover it is confusing to me that fisheries are both a service and a stressor, this should be clarified.

Response: We have clarified this by explicitly pointing out that fisheries can be treated as both ecosystem services and stressors, because fisheries harvest fish biomass for food, but fisheries also damage underlying ecosystems and can reduce access and quality of ecosystem services. The existence of fisheries as both ecosystem service and stressor/impacting activity is established in the literature (Raymond et al. 2013, Singh et al. 2017). When we refer to fisheries as ecosystem services we are actually recognizing the “benefits from fisheries” as the benefits that humans derive from ecosystems through fisheries harvest. When we refer to fisheries as stressor, we are recognizing the impacts to ecosystem services that fisheries generate. We have referred to “benefits from fisheries” throughout the text now, to help clarify the difference. We have added the following text (line 237-244): 

Many human activities, such as fishing, access benefits from ecological processes and play important roles in ecosystem service delivery to people while also contributing impacts towards ecosystem services [30]. We treat these activities (e.g. fishing), therefore as both ecosystem services as well as drivers that cause impact (following Singh et al. 2017). To distinguish between these multiple roles that fisheries play, we emphasize benefits when labeling fisheries as ecosystem services (such as “benefits from commercial demersal fisheries”) and emphasize impacts when labeling fisheries as drivers of impact (such as “demersal destructive fishing”). 

Raymond CM, Singh GG, Benessaiah K, Bernhardt JR, Levine J, et al. (2013) Ecosystem services and beyond: Using multiple metaphors to understand human–environment relationships. BioScience 63: 536-546.

Singh GG, Sinner J, Ellis J, Kandlikar M, Halpern BS, et al. (2017) Mechanisms and risk of cumulative impacts to coastal ecosystem services: An expert elicitation approach. Journal of environmental management 199: 229-241.

Reviewer: Lines 250-253: this should be in some way explained before when the concept of cumulative impact is introduced.

Response: Since we have now defined cumulative impacts, we hope this helps clarify this point. We have also added the following to the methodological overview (lines 187-190):

we overlaid maps of drivers of impact (with impact scores) on maps of ecosystem service to assess the cumulative impacts of all available activities on each service, in accordance with our definition of cumulative impacts.

Reviewer: 

Results

In my opinion results should be simplified and better presented

First, authors should avoid to list indicator values in text as list, please use table. Moreover presenting impact scores were calculated before without the service and value dimensions and later with service and value dimensions makes difficult to read the paper. I suggest to remove this or to present it in clearer way, for example with two separate sessions. Moreover as I already told authors should better explain in method what are these “dimensions”: supply dimensions, values dimensions...there are too many concepts and terms and this confusing. As I already suggest in the first review is necessary to focus on some main results they want to convey and present them in a clearer and synthetic way.

Response: As clarified by the reviewer on further communication, we have removed the in-text results and instead pointed to the relevant figures which showcase the results. We have also added a table which shows all the impact scores and refer to this table (Table 2). We have also further explained the supply, service, and value dimensions in the introduction as demonstrated in an earlier comment. Finally, we have separated results of ES supply service and value and results of just ES supply by incorporating another header level.

Reviewer: Moreover I think that the part regarding InVEST, mapping and the cumulative index should be clearly separated in both methods and results and merged when put together to make the cumulative impact. I think this should be a good framework for the entire paper:

-maps

- expert

-cumulative measures

Response: We have considered this suggestion but cannot fully implement it because our results do not separate the InVEST mapping and the cumulative impact scores. The cumulative impact mapping framework that we adopt here (from Halpern et al 2008, 2009, 2015) does not separate the maps, expert values and cumulative impacts but synthesizes them all to produce maps of impact. 

However, we have ensured that our methods are largely put together in the sequence that the reviewer suggests. We agree with the reviewer that this structure is useful for the methods. We have moved the methods sections on future risk and impact mechanisms to follow the expert elicitation for risk scores because these are also expert derived.

Halpern BS, Walbridge S, Selkoe KA, Kappel CV, Micheli F, et al. (2008) A Global Map of Human Impact on Marine Ecosystems. Science 319: 948-952.

Halpern BS, Kappel CV, Selkoe KA, Micheli F, Ebert CM, et al. (2009) Mapping cumulative human impacts to California Current marine ecosystems. Conservation letters 2: 138-148.

Halpern BS, Frazier M, Potapenko J, Casey KS, Koenig K, et al. (2015) Spatial and temporal changes in cumulative human impacts on the world’s ocean. Nature communications 6: 1-7.

Reviewer: Lines 350-355 should be put in a table

Response: We have pointed to the relevant figure, as well as the new Table 2 which shows these values.

Reviewer: Line 355: is the supply dimension identified by identified by spatial representation of ecosystem services? Please be consistent: it must be clear to which part of the methods the authors refer in results

Response: Since we have further clarified what ecosystem service supply, service, and value dimensions are in the introduction and methods, we believe we have clarified this further. The supply dimension is looking at impact only considering impact to the biophysical system that produces the ecosystem service and not the service and value dimensions that consume the service. We have also further clarified this line to read (line 385-388): “When mapping the per-cell cumulative impact model while only considering impacts to ecosystem service supply (and not including service and value dimensions), the ranked list of ecosystem services facing impacts is similar to the list considering service and value dimensions, with some differences.”

Reviewer: Lines 357-381: need table with Ic

Response: We have pointed to the relevant figure and table for the Ic values in text. Note that lines 362-381 (now 394-417) are figure captions and so implicitly refer to the figures.

Reviewer: Line 382: how is it calculated the “total summed impact”, insert also these values in a table. It is not necessary to list all values in the text and this makes the reading very hard. If values are diffent for only supply dimension and not please insert both

Response: We understand this term can be confusing. To be more consistent, we have changed this to “cumulative impact” and refer to the calculation of Ic. We have also removed the in-text results and pointed to the relevant figures (and Table 2) which show these data.

Reviewer: Lines 391:where can I see this? Fig. 1? If yes please cite here

Response: We have cited the figures now

Reviewer: Lines 421-430: again, these results should be presented with a table, directly compared with the previous ones. Please name the different Ic with different names: the reader reads all along the text Ic but if I correctly understand this sometimes refers to the index with value dimension, sometimes not.

Response: We have removed the in-text results here and put them in a table for both the per-pixel and total Ic values (Ic across spatial range), since no figure clearly shows the proportional change in the impact scores when considering ecosystem service service and value dimensions on top of supply dimensions. We have also labeled the per-pixel Ic values and the total Ic values across the spatial range to differentiate them. We have also ensured that when we refer to Ic values that include service and value dimensions we indicate this, and when we refer to Ic values that do not include service and value dimensions we refer to Ic values for supply dimensions.

Reviewer: Figure 6: it should be better explained in methods how these values are obtained

Response: We have rewritten the section in the methods to describe how we gathered this data from experts. The section now reads (lines 340-352): 

We asked experts in the risk survey to indicate whether or not the given drivers of impact affected their chosen ecosystem service directly or indirectly (or neither or both), with an optional follow-up to describe the mechanism of impact. Each driver of impact were grouped in one of four different categories: fisheries impacts, coastal commercial impacts, land-based impacts, and climate change impacts. Fisheries impacts includes all those drivers related to fisheries including demersal destructive and non-destructive fishing, pelagic fishing, and recreational fishing. Coastal commercial impacts include coastal industries such as aquaculture, shipping, ports, docks, log dumping, ocean dumping. Land-based impacts include industry, pulp and paper, onshore mining, human settlements, forestry, and agriculture. Climate change impacts include ocean acidification, sea level rise, sea temperature change, and UV change. When all drivers were categorized, we calculated the proportion of direct versus indirect impacts (also accounting for impacts that could be both or neither) within each category affect each ecosystem service.

---

## [Editor Report · Decision Letter 2]

3 Mar 2020

Mapping cumulative impacts to coastal ecosystem services in British Columbia

PONE-D-19-18924R2

Dear Dr. Singh,

We are pleased to inform you that your manuscript has been judged scientifically suitable for publication and will be formally accepted for publication once it complies with all outstanding technical requirements.

With kind regards,

Carlo Nike Bianchi

Academic Editor

PLOS ONE
---

## [Editor Report · Acceptance letter]

24 Apr 2020

PONE-D-19-18924R2 

Mapping cumulative impacts to coastal ecosystem services in British Columbia 

Dear Dr. Singh:

I am pleased to inform you that your manuscript has been deemed suitable for publication in PLOS ONE. Congratulations! Your manuscript is now with our production department. 

With kind regards,

on behalf of

Dr. Carlo Nike Bianchi 

Academic Editor

PLOS ONE